# Structure and mechanism of a Hypr GGDEF enzyme that activates cGAMP signaling to control extracellular metal respiration

Zachary F Hallberg[1†‡], Chi Ho Chan[2†], Todd A Wright[1], Philip J Kranzusch[3,4,5], Kevin W Doxzen[6§], James J Park[1], Daniel R Bond[2*], Ming C Hammond[1,7,8#¶*]

[1]Department of Chemistry, University of California, Berkeley, Berkeley, United States; [2]Department of Plant and Microbial Biology and BioTechnology Institute, University of Minnesota, Minnesota, United States; [3]Department of Microbiology and Immunobiology, Harvard Medical School, Boston, United States; [4]Department of Cancer Immunology and Virology, Dana-Farber Cancer Institute, Boston, United States; [5]Parker Institute for Cancer Immunotherapy at Dana-Farber Cancer Institute, Boston, United States; [6]Biophysics Graduate Group, University of California, Berkeley, Berkeley, United States; [7]Department of Molecular & Cell Biology, University of California, Berkeley, Berkeley, United States; [8]Department of Chemistry, University of Utah, Salt Lake City, United States

*For correspondence:
dbond@umn.edu (DRB);
mingch@chem.utah.edu (MCH)

[†]These authors contributed equally to this work

Present address: [‡]Department of Plant and Microbial Biology, University of California, Berkeley, Berkeley, United States; [§]Innovative Genomics Institute, University of California, Berkeley, Berkeley, United States; [#]Henry Eyring Center for Cell & Genome Science, University of Utah, Salt Lake City, United States; [¶]Department of Chemistry, University of Utah, Salt Lake City, United States

**Abstract** A newfound signaling pathway employs a GGDEF enzyme with unique activity compared to the majority of homologs associated with bacterial cyclic di-GMP signaling. This system provides a rare opportunity to study how signaling proteins natively gain distinct function. Using genetic knockouts, riboswitch reporters, and RNA-Seq, we show that GacA, the Hypr GGDEF in *Geobacter sulfurreducens*, specifically regulates cyclic GMP-AMP (3′,3′-cGAMP) levels in vivo to stimulate gene expression associated with metal reduction separate from electricity production. To reconcile these in vivo findings with prior in vitro results that showed GacA was promiscuous, we developed a full kinetic model combining experimental data and mathematical modeling to reveal mechanisms that contribute to in vivo specificity. A 1.4 Å-resolution crystal structure of the *Geobacter* Hypr GGDEF domain was determined to understand the molecular basis for those mechanisms, including key cross-dimer interactions. Together these results demonstrate that specific signaling can result from a promiscuous enzyme.
DOI: https://doi.org/10.7554/eLife.43959.001

## Introduction

A cell's sensory system is composed of complex signaling networks that permit timely responses to changes in environmental conditions, cues from neighboring cells, and feedback from contacting surfaces (*Camilli and Bassler, 2006*; *Capra and Laub, 2012*). How new signaling pathways emerge to control distinct functions remains an important underlying question (*Rowland and Deeds, 2014*). While highly conserved signaling enzymes are generally easy to identify at the sequence level, it is challenging to predict their specific activity or role (*Danchin et al., 2018*; *Seshasayee et al., 2010*). Another complication is that signaling enzymes often exhibit promiscuous or off-target activity when studied in vitro (*Rowland and Deeds, 2014*). In vivo confirmation of both their product and physiological function is essential.

**eLife digest** Microscopic organisms known as bacteria are found in virtually every environment on the planet. One reason bacteria are so successful is that they are able to form communities known as biofilms on surfaces in animals and other living things, as well as on rocks and other features in the environment. These biofilms protect the bacteria from fluctuations in the environment and toxins.

For over 30 years, a class of enzymes called the GGDEF enzymes were thought to make a single signal known as cyclic di-GMP that regulates the formation of biofilms. However, in 2016, a team of researchers reported that some GGDEF enzymes, including one from a bacterium called *Geobacter sulfurreducens*, were also able to produce two other signals known as cGAMP and cyclic di-AMP. The experiments involved making the enzymes and testing their activity outside the cell. Therefore, it remained unclear whether these enzymes (dubbed 'Hypr' GGDEF enzymes) actually produce all three signals inside cells and play a role in forming bacterial biofilms.

*G. sulfurreducens* is unusual because it is able to grow on metallic minerals or electrodes to generate electrical energy. As part of a community of microorganisms, they help break down pollutants in contaminated areas and can generate electricity from wastewater. Now, Hallberg, Chan et al. – including many of the researchers involved in the 2016 work – combined several experimental and mathematical approaches to study the Hypr GGDEF enzymes in *G. sulfurreducens*.

The experiments show that the Hypr GGDEF enzymes produced cGAMP, but not the other two signals, inside the cells. This cGAMP regulated the ability of *G. sulfurreducens* to grow by extracting electrical energy from the metallic minerals, which appears to be a new, biofilm-less lifestyle. Further experiments revealed how Hypr GGDEF enzymes have evolved to preferentially make cGAMP over the other two signals.

Together, these findings demonstrate that enzymes with the ability to make several different signals, are capable of generating specific responses in bacterial cells. By understanding how bacteria make decisions, it may be possible to change their behaviors. The findings of Hallberg, Chan et al. help to identify the signaling pathways involved in this decision-making and provide new tools to study them in the future.

DOI: https://doi.org/10.7554/eLife.43959.002

In the domain Bacteria, the second messenger signaling pathway involving cyclic di-GMP (cdiG) is used almost universally to shift between a free-living and surface-attached biofilm state, which requires coordinated changes in physiology and gene regulation (*Jenal et al., 2017*; *Römling et al., 2013*). Multiple processes such as flagellar motility, exopolysaccharide production, quorum sensing, and pilus retraction are controlled by cdiG (*Hengge, 2009*). In pathogenic bacteria, cdiG regulates additional virulence factor secretion, host suppression, and defense mechanisms (*Chen et al., 2014*; *Valentini and Filloux, 2016*). These processes are each driven by specific sets of signaling enzymes: diguanylate cyclases harboring GGDEF domains and EAL/HD-GYP domain phosphodiesterases, which respectively synthesize or degrade cdiG in response to sensory modules fused directly to the enzyme or acting upstream.

A grand challenge in bacterial signaling is to understand how cdiG networks utilize only one intra-cellular output to control diverse adaptations. While many explanations to the 'one signal, many phenotypes' problem have been explored (*Hobley et al., 2012*; *Sarenko et al., 2017*; *Hug et al., 2017*), we recently discovered an unexpected alternative that involves a new activity for GGDEF enzymes. A sub-class of GGDEFs demonstrates promiscuous activity and is capable of producing all three known bacterial cyclic dinucleotides, cdiG, cyclic di-AMP (cdiA), and cyclic GMP-AMP (3′,3′-cGAMP, also called cyclic AMP-GMP) (*Hallberg et al., 2016*). The hybrid promiscuous (Hypr) GGDEF enzyme GSU1658 (renamed GacA for GMP-AMP cyclase) was hypothesized to regulate a new signaling pathway through activation of cGAMP-specific GEMM-Ib riboswitches in *Geobacter sulfurreducens*, an environmental bacterium known for its unique ability to perform extracellular electron transfer and accelerate bioremediation of subsurface contaminants (*Kellenberger et al., 2015*; *Nelson et al., 2015*). However, the physiological function of cGAMP signaling was not established, and it was unclear whether GacA regulated a single (only cGAMP) or multiple cyclic dinucleotide

signaling pathways in vivo. Furthermore, there were general questions about the molecular mechanism of homodimeric GGDEF enzymes, the most abundant class of cyclic dinucleotide signaling enzymes in bacteria (*Seshasayee et al., 2010*), such as the function of highly conserved residues (*Schirmer, 2016*) and the identity of the general base, as well as specific questions for how Hypr variants could perform preferential synthesis of cGAMP, a heterodimeric product.

In this work, we show for the first time that GacA specifically affects cGAMP levels and cGAMP riboswitch transcripts in *G. sulfurreducens* and is important during bacterial growth on particulate acceptors, such as mineral Fe(III) oxides. In contrast, GacA is not essential for biofilm growth on electrodes, a phenotype associated with cdiG signaling. These results reveal that the general physiological function for cGAMP is to establish a transiently attached lifestyle that is distinct from the permanently attached biofilm lifestyle signaled by cdiG.

Furthermore, we sought to understand the molecular mechanism for GacA by obtaining a 1.4 Å-resolution structure of the *Geobacter* Hypr GGDEF domain bound to GTP. Combining this structure with kinetic analyses and mathematical modeling afforded insights into how GGDEF enzymatic activity is regulated in general as well as uncovered natural variations that give rise to cGAMP synthesis. Together, these genetic and biochemical analyses provide evidence that this cGAMP signaling pathway emerged from components of cdiG signaling to regulate a distinct surface-associated lifestyle, and gives a full picture of cGAMP signaling from the molecular to the cellular to the environmental level.

## Results and discussion

### GacA is necessary for Fe(III) particle-associated growth, but not electricity production at electrode surfaces

*Geobacter* isolates produce energy via contact-dependent electron transfer to extracellular metals, which exist as insoluble precipitates at neutral pH (*Navrotsky et al., 2008*). Stimulation of this biological metal reduction activity is useful for bioremediation of metal-rich sites and anaerobic oxidation of petroleum-based groundwater pollutants (*Chang et al., 2005*; *Lovley et al., 2011*; *Rooney-Varga et al., 1999*). *Geobacter* also can grow via electron transfer to electrode surfaces, where their biofilms produce electricity in bioelectrochemical devices that use wastewater or contaminated groundwater (*Bond and Lovley, 2003*; *Logan and Rabaey, 2012*; *Lovley, 2012*). The ability to transfer electrons to extracellular substrates requires multiple extracellular structures, including pili, polysaccharides, and cytochromes localized to the outer cell surface.

cGAMP-responsive riboswitches (GEMM-Ib family) are conserved upstream of many cytochrome, pilus assembly, and polysaccharide biosynthesis genes in most *Geobacter* species (*Kellenberger et al., 2015*; *Nelson et al., 2015*), suggesting a possible role for this cyclic dinucleotide in attachment to extracellular surfaces that serve as electron acceptors. The discovery of a GMP-AMP cyclase in *Geobacter* (GacA) (*Hallberg et al., 2016*) presented the hypothesis that GacA synthesizes cGAMP in vivo to alter gene expression via cGAMP-specific riboswitches. However, no Hypr GGDEF enzyme including GacA has been linked yet to intracellular cGAMP levels or to phenotypic changes in any organism.

To assess the physiological role of cGAMP signaling in *G. sulfurreducens*, we constructed a scarless *gacA* deletion strain and tested its ability to respire soluble and particulate extracellular electron acceptors (See Key Resources Table). The Δ*gacA* strain was defective in reducing Fe(III) oxide particles, including both akaganeite and amorphous insoluble Fe(III)-(oxyhydr)oxide, but grew normally with soluble compounds such as fumarate or Fe(III)-citrate (*Figure 1A and B*). Mutants lacking *gacA* always demonstrated a ~5 d lag during reduction of Fe(III) oxides, but if left exposed for over 14 d, the Δ*gacA* strain eventually produced Fe(II) at levels similar to wild type. Re-expressing *gacA* as a single copy on the chromosome restored Fe(III) reduction, and caused it to initiate sooner than wild type. In contrast, after a ~1 day lag, the rate and extent of growth as a biofilm attached to electrodes poised at −0.1 V vs SHE (a potential chosen to mimic Fe(III) oxides) was unaffected by deletion of *gacA* (*Figure 1C*).

The Δ*gacA* defect with Fe(III) oxides was the opposite of mutants in *esn* genes encoding chemosensory, histidine kinase, and diguanylate cyclase response regulator proteins. Mutants in *esn* genes reduce Fe(III) oxides similar to wild type, but show poor biofilm growth on electrodes (*Chan et al.,*

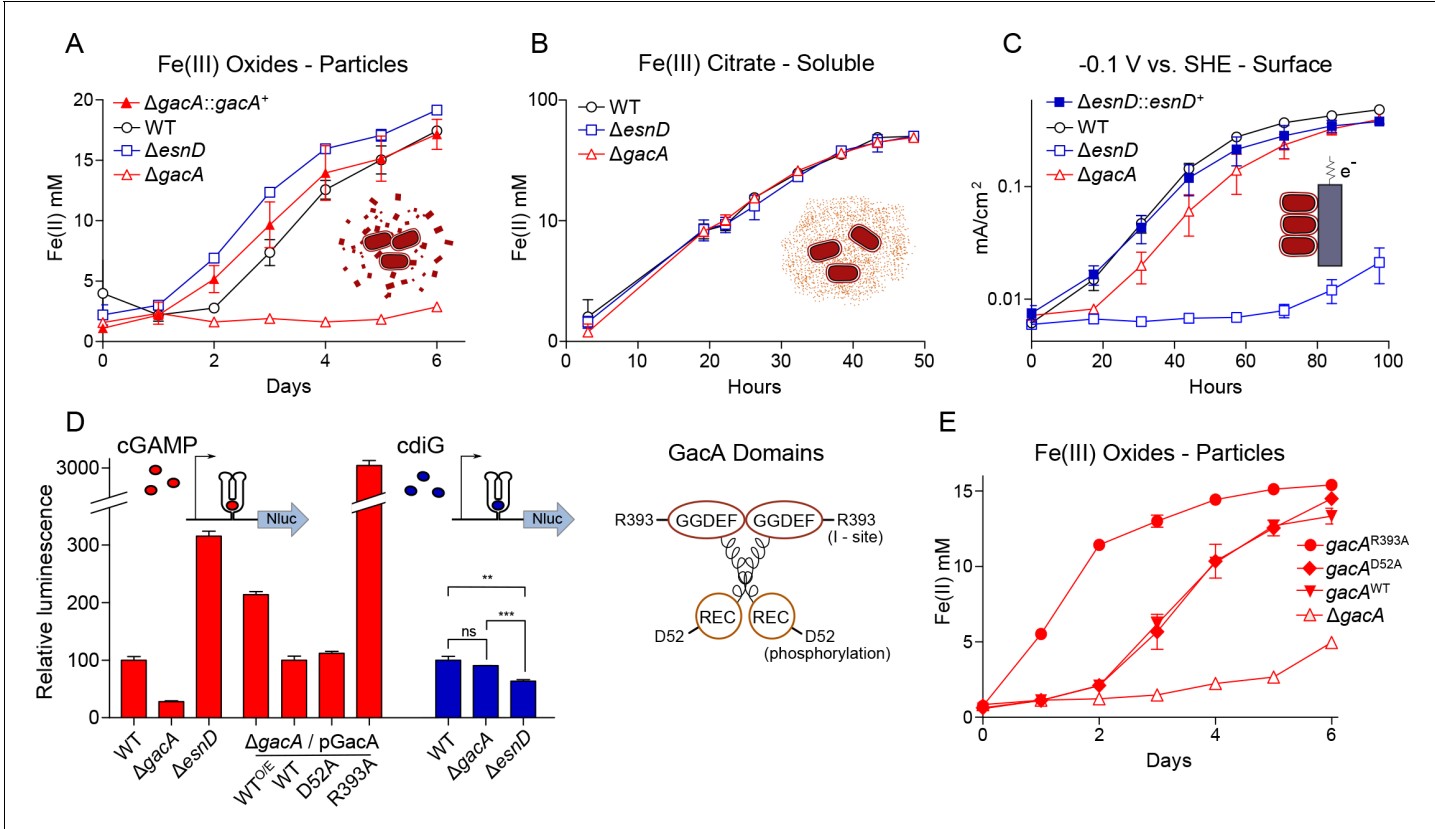

**Figure 1.** GacA synthesizes cyclic GMP-AMP and controls Fe(III) particle respiration in vivo, whereas EsnD synthesizes cyclic di-GMP and controls electrode respiration. (**A**) *G. sulfurreducens* Δ*gacA* deletion strain is defective in the reduction of insoluble Fe(III) oxide particles (open triangles). Re-expressing *gacA* in the Tn7 site on the chromosome (filled triangles) restores Fe(III) reduction. An Δ*esnD* deletion strain (squares) exhibits no lag in Fe(III) oxide reduction compared to 2 d after inoculation for WT (circles). (**B**) Soluble Fe(III) citrate reduction is unaffected in either Δ*gacA* (triangles) and Δ*esnD* (squares) strains relative to WT (circles). (**C**) EsnD is required for robust electrode reduction (open squares) and re-expressing *esnD* in the Tn7 site on the chromosome (filled squares) restores electrode respiration to WT (circles) levels. GacA is not required for electrode respiration (triangles). (**D**) Strains containing riboswitches driving Nanoluc luciferase, reporting cellular levels of cGAMP (red bars) and cdiG (blue bars). The Δ*gacA* strains express GacA variants from the Tn7 chromosomal site (also see cartoon). Deleting *gacA* reduces cGAMP-dependent reporter levels by ~80% but has no effect on cdiG reporter levels. Deleting *esnD* reduces cdiG levels by ~20% and increases cGAMP levels to 3x WT. The Rec domain variant D52A produces cGAMP at levels comparable to WT, while the I-site variant R393A increases cGAMP levels to 30x WT. (**E**) Reduction of Fe(III) oxide particles in the Δ*gacA* deletion strain is rescued by re-expressing WT GacA and D52A variants, and exhibits significantly increased rate for the R393A variant that overproduces cGAMP. Representative biological replicates are shown for Fe(III) oxide reduction (n = 3), Fe(III) citrate reduction (n = 3), and electrode growth (n = 4). Nanoluc assays were performed in biological replicates (n = 3) in panel D. P values in panel D: ns > 0.05; **=0.001–0.01; ***<0.001. All error bars represent standard deviations.

DOI: https://doi.org/10.7554/eLife.43959.003

The following source data is available for figure 1:

**Source data 1.** Growth rates and nanoluc reporter data.
DOI: https://doi.org/10.7554/eLife.43959.004

*2017*). One of these *esn* genes, *esnD* (GSU3376), encodes a GGDEF diguanylate cyclase that produces only cdiG based on biosensor analysis (*Hallberg et al., 2016*). We compared growth of a mutant lacking cdiG-producing EsnD under the same conditions used to study the mutant lacking cGAMP-producing GacA. The Δ*esnD* mutant reduced Fe(III) oxides more rapidly than wild type, but showed a > 3 d lag in colonizing electrode surfaces and never reached current levels observed in wild type (*Figure 1A and C*). Re-expressing *esnD* as a single copy integrated into the chromosome restored biofilm growth on −0.1 V vs. SHE electrodes (*Figure 1C*). These data support that cdiG contributes to biofilm growth on electrodes, while cGAMP is involved in reduction of Fe(III) oxide particles. In addition, the enhanced growth of Δ*esnD* mutants with Fe(III) oxides suggests an antagonistic effect of cyclic di-GMP on Fe(III) oxide reduction.

## GacA is essential for production of intracellular cyclic GMP-AMP (cGAMP)

To test if deletion of *gacA* altered cGAMP levels within the cell, we developed a new luciferase-based reporter by cloning the cGAMP-specific *pgcA* riboswitch upstream of a nano-luciferase (NLuc) gene, then integrating this reporter as a single copy into the Tn7 insertion site of *G. sulfurreducens*. For these experiments, all constructs were grown under the same conditions to stationary phase under electron acceptor limitation. Luminescence in the cGAMP reporter strain declined over 80% when *gacA* was deleted, and recovered to wild type luminescence levels when *gacA* was re-expressed from its native promoter (*Figures 1D*, *2A and B*). Over-expression of *gacA* from a constitutive promoter increased luminescence to 200% of wild type. This result links *gacA* to intracellular levels of cGAMP in *Geobacter*, and correlates with parallel LC/MS analysis of cell extracts, which also showed that cGAMP levels fell below the detection limit in the Δ*gacA* strain (*Figure 2C*).

The new cGAMP reporter assay also allowed us to interrogate roles of conserved residues in GacA that are critical for activity in diguanylate cyclases. GacA has an N-terminal CheY-like receiver domain and a C-terminal GGDEF domain. In WspR, a *Pseudomonas aeruginosa* diguanylate cyclase with similar domain architecture, phosphorylation of a conserved aspartate in the receiver domain promotes dimerization, leading to an active enzyme (*De et al., 2008*; *Huangyutitham et al., 2013*). In a *G. sulfurreducens* Δ*gacA* background, expression of a GacA$^{D52A}$ variant with the phosphorylation site replaced by alanine produced similar levels of cGAMP as GacA$^{WT}$. Expression of *gacA*$^{D52A}$ also restored Fe(III) oxide reduction to a Δ*gacA* mutant (*Figure 1D and E*). Thus, it appears that the conserved aspartate is not essential for GacA activation. Aspartate phosphorylation instead may deactivate GacA or the receiver domain may be activated through non-canonical mechanisms (*Lin et al., 2009*; *Ocasio et al., 2015*; *Trajtenberg et al., 2014*; *Wang et al., 2009*).

A second mechanism regulating canonical GGDEF domains involves a conserved allosteric inhibitory site (I-site) that binds cyclic dinucleotides. We previously showed that WT GacA co-purifies predominantly with cdiG bound and has low activity when expressed in *E. coli*, whereas the R393A I-site mutant of GacA does not purify with bound dinucleotides and exhibits increased in vitro activity (*Hallberg et al., 2016*). In extracts of cells overexpressing GacA, the major CDN present is cGAMP (*Hallberg et al., 2016*), yet GacA still purifies with bound cdiG, which supports the hypothesis that the I-site is specific for cdiG. When we expressed the GacA$^{R393A}$ variant insensitive to allosteric inhibition in a Δ*gacA* background, cGAMP-dependent luciferase reporter activity increased 30-fold compared to wild type. Expressing this highly active I-site mutant in the Δ*gacA* strain also led to the highest observed rates of Fe(III) oxide reduction, nearly doubling the level of Fe(II) produced at all time points (*Figure 1D and E*). These data suggest that occupancy of the I-site, rather than phosphorylation of the receiver domain, primarily regulates GacA activity. The increased Fe(III) reduction activity of the cGAMP-overproducing strain further supports a role for cGAMP in enhancing metal oxide reduction.

The Δ*esnD* strain also reduced amorphous insoluble Fe(III)-oxides more rapidly than the wild type strain (*Figure 1A*), leading to a hypothesis that deletion of *esnD* increased cGAMP levels in *G. sulfurreducens*. Consistent with this hypothesis, activity of the cGAMP reporter was 3-fold higher when *esnD* was deleted (*Figure 1D*). Using a different reporter construct comprised of an engineered cdiG-responsive riboswitch (*Figures 1D*, *2A and B*), we confirmed that deletion of *esnD* caused a detectable decrease in cdiG, while cdiG levels did not change significantly in Δ*gacA*. These results support cGAMP synthesis by GacA being inhibited by intracellular cdiG levels. Mutation of the I-site appears to relieve this allosteric inhibition, as does lowering of cdiG levels in the Δ*esnD* strain, triggering corresponding increases in Fe(III) oxide reduction rates.

## The global transcriptional response to altered cGAMP levels is focused on riboswitch regulons

Many GEMM-I riboswitches from *Geobacter* selectively bind cGAMP over other cyclic dinucleotides such as cdiG and are the founding members of the GEMM-Ib subclass. For example, in-line probing showed the riboswitch upstream of *pgcA* used for our reporter analysis was ~1200 fold more selective for cGAMP over cdiG (*Kellenberger et al., 2015*), and similar results were reported in (*Nelson et al., 2015*). The correlation we observed between increased Fe(III) oxide reduction and

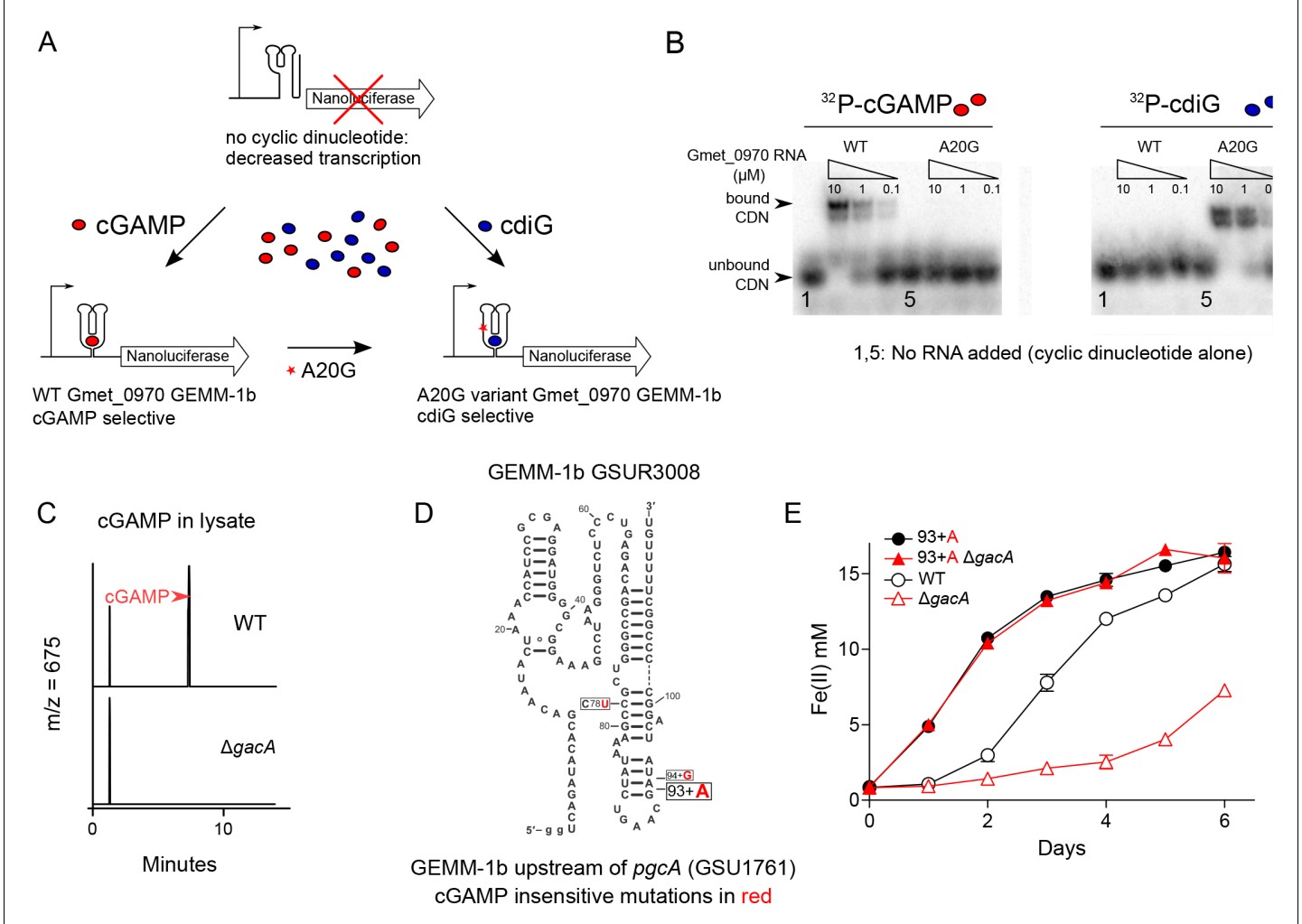

**Figure 2.** Nanoluciferase fusion to WT cGAMP riboswitch and engineered cdiG riboswitch for in vivo cyclic dinucleotide analysis; riboswitch GSUR3008 variants that control *pgcA* (GSU1761) expression in *Geobacter sulfurreducens*. (**A**) Expression of nanoluciferase is induced by binding of cyclic dinucleotides to GEMM-1b riboswitch. An A to G variant in nucleotide 20 in the Gmet_0970 GEMM-1b riboswitch changes cGAMP selectivity to cdiG. (**B**) The WT and A20G variant of Gmet_0970 GEMM-1b riboswitch are selective for cGAMP and cdiG over a 100-fold range of riboswitch RNA concentration, respectively. (**C**) LC-MS chromatogram traces for the ion extraction of cGAMP (m/z = 675) from cell lysates for *G. sulfurreducens* PCA WT or Δ*gacA* strains. (**D**) Single nucleotide polymorphisms (SNP) highlighted in red in GSUR3008 that cause cGAMP insensitive variants to demonstrate constitutive expression of pgcA (GSU1761). The C78U and 94 + G variants were isolated in a previous study selecting for increased rate of Fe(III) oxide reduction in *G. sulfurreducens* (**Tremblay et al., 2011**). The cGAMP insensitive C78U variant was further confirmed by in-line probing (**Kellenberger et al., 2015**). The cGAMP insensitive 93 + A variant is described in this study. (**E**) The GSUR3008 93 + A mutation in *G. sulfurreducens* eliminates the requirement for GacA or cGAMP for Fe(III) oxide reduction due to the over-expression of *pgcA* that was confirmed by RNAseq. No other gene expression changes were detected. Raw reads are deposited in the NCBI SRA database PRJNA290373.

DOI: https://doi.org/10.7554/eLife.43959.005

increased cGAMP levels, combined with the in vitro activity of cGAMP riboswitches, suggests that cGAMP could be a global effector of genes crucial to metal reduction.

There are 17 GEMM-I riboswitches in *G. sulfurreducens*, and in several cases, two riboswitches occur in tandem upstream of a gene or operon. When *gacA* was deleted, RNAseq analysis showed that transcription of all genes downstream of GEMM-lb riboswitches declined (**Figure 3A**, **Supplementary file 1**). For example, deletion of *gacA* decreased expression of an operon containing outer membrane cytochromes OmcAHG (GSU2885-GSU2882, 16-fold decrease), an operon of uncharacterized lipoprotein transpeptidases (GSU0181-0183, 16-fold decrease), the extracellular cytochrome PgcA (10-fold decrease), a transcriptional regulator and genes within the *pilMNOP*

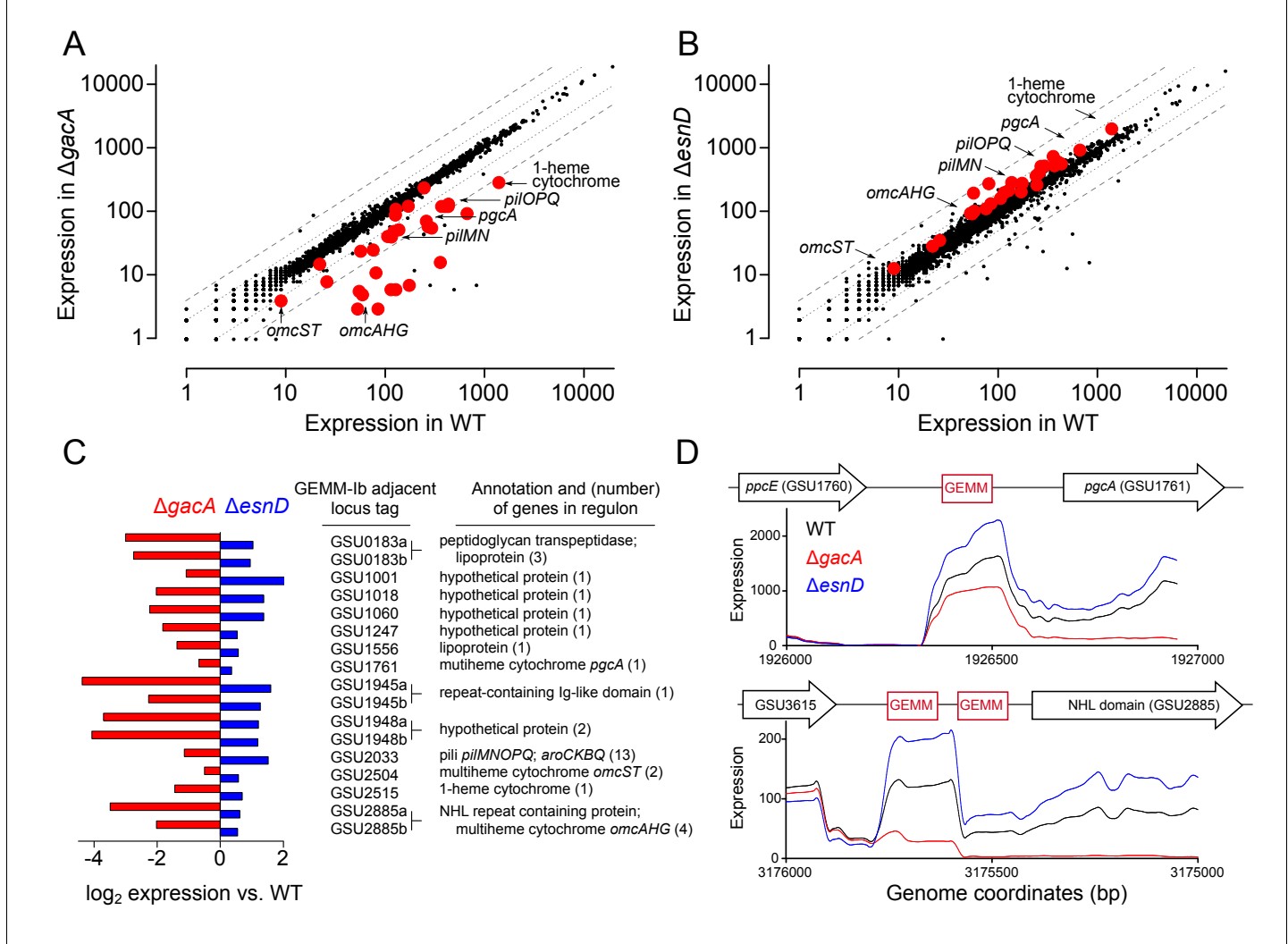

**Figure 3.** GEMM-1b riboswitch-controlled genes are downregulated in a Δ*gacA* deletion strain and upregulated in a Δ*esnD* deletion strain. Full RNAseq expression data are in ***Supplementary file 1*** (**A**) Comparison of all genes expressed in WT vs Δ*gacA* deletion strain. Each dot represents expression of a gene; red dots highlight GEMM-1b regulated genes. Dotted and dashed lines indicate 2- and 4-fold expression differences. Most GEMM-1b regulated genes have at least 2–4 fold decrease in expression compared to WT. (**B**) Comparison of all genes expressed in WT vs. a Δ*esnD* deletion strain, labeled as in part **A**. Most GEMM-1b regulated genes have at least a 2-fold increase in expression compared to WT. (**C**) GEMM-1b riboswitch transcripts are decreased in a Δ*gacA* deletion and increased in a Δ*esnD* deletion compared to WT. GEMM-1b sequences are listed by the first gene locus tag they regulate. Some genes have tandem GEMM-1b sequences upstream, these are labeled by their locus tags followed by 'a' and 'b'. (**D**) RNAseq reads from WT, Δ*gacA*, and Δ*esnD* mapped to the upstream region of GSU1761 (top) and GSU2885 (bottom). GSU2885 is an example of a gene that is regulated by tandem riboswitches. Genes are not drawn to scale with the riboswitch sequences. RNAseq data are generated from biological replicates (n = 2).

DOI: https://doi.org/10.7554/eLife.43959.006

The following source data is available for figure 3:

**Source data 1.** Geobacter mRNA expression levels.
DOI: https://doi.org/10.7554/eLife.43959.007

operon (2–3 fold decrease), the extracellular multiheme cytochromes OmcST, (2–3 fold decrease), and multiple hypothetical genes (***Figure 3A***).

In contrast, cells lacking *esnD* showed a ~2 fold increase in expression of these same genes controlled by cGAMP-responsive riboswitches (***Figure 3B***). Every gene that showed a decrease in expression due to *gacA* deletion also showed an increase due to Δ*esnD*. The smallest effect was in the operon containing OmcST, which was previously reported to contain a riboswitch sensitive to

both cGAMP and cdiG (*Kellenberger et al., 2015*). Interestingly, some hypothetical genes (GSU0919, GSU3250, and GSU3409), and the entire operon for the multiheme cytochrome OmcZ were downregulated in both ΔgacA and ΔesnD despite a lack of known riboswitch sequences, suggesting additional modes for cyclic dinucleotide regulation. As OmcZ is known to be essential for growth in electrode biofilms (*Nevin et al., 2009*), the unexpected decrease in OmcZ due to *gacA* deletion likely explains the lag in electrode growth seen in *Figure 1C*. This effect may be due to minor contribution by GacA to cdiG levels or from pleiotropic effects from cGAMP signaling or its many downstream effectors.

Closer inspection of intergenic regions confirmed lower riboswitch mRNA levels and increased termination near the cGAMP recognition site in ΔgacA mutants. A table showing the inverse relationship in riboswitch RNA levels between ΔgacA and ΔesnD strains is shown in *Figure 3C* and a map of two different untranslated regions is shown in *Figure 3D*. For example, in the tandem riboswitch upstream of the OmcAHG gene cluster, deletion of *gacA* eliminated detectable RNA by the second riboswitch sequence, while RNA levels in each riboswitch region increased ~2 fold in the ΔesnD strain.

Previously published experiments selecting for faster Fe(III) oxide reduction activity resulted in evolved *G. sulfurreducens* variants containing mutations in the *pgcA* GEMM-Ib riboswitch (GSUR3008) (*Tremblay et al., 2011*). Similarly, we identified a naturally evolved variant with a single A inserted after residue 93 of the *pgcA* riboswitch that had accelerated growth under laboratory conditions in which Fe(III) oxides were abundant. The A insertion is predicted to destabilize the riboswitch terminator, which in another terminator mutant we showed bypasses the need for cGAMP binding to turn on gene expression (*Kellenberger et al., 2015*). To test if faster Fe(III) reduction could be explained by cells becoming insensitive to cGAMP regulation, a scarless and markerless *G. sulfurreducens* strain was made to reconstruct this natural variation in a clean genetic background (*Figure 2E*). The Fe(III) oxide reduction rate was indeed increased in this $93^{+A}$ strain. When *gacA* was deleted in this $93^{+A}$ strain, *pgcA* expression and metal reduction rate remained high. Global transcriptional analysis showed that only expression of *pgcA* was increased, and was insensitive to cGAMP signaling, with no other cGAMP-dependent genes, genes for other electron transfer proteins, or genes for pili affected (data not shown).

These genetic and RNAseq experiments establish a unique phenotype controlled by 3'3'-cGAMP, and show that GacA is primarily responsible for formation of this second messenger. An opposing relationship for cGAMP and cdiG is established, with each cyclic dinucleotide enhancing extracellular electron transfer to a distinct type of surface. To our knowledge, this is the first time that these two contact-dependent electron transfer processes have been shown to be differentially regulated on a global scale. In this context, the signaling enzyme GacA presents an enigma: we previously discovered that this founding member of Hypr GGDEFs can produce mixtures of cdiG, cGAMP, and cdiA, depending on in vitro conditions (*Hallberg et al., 2016*). In the next sections, we employ biochemical analysis, mathematical modeling, and structural elucidation to determine how this homodimeric enzyme 'breaks symmetry' in several ways to produce the heterodimeric cGAMP signal in the cell. In addition, the role of several residues ultra-conserved across all GGDEF enzymes but with previously unassigned function is revealed.

## GacA differs in mechanism from DncV and cGAS

To date, two other enzymes have been discovered that produce cGAMP or the related compound, mixed linkage cGAMP: DncV from *Vibrio cholerae*, and cGAS in metazoans (*Davies et al., 2012*; *Sun et al., 2013*; *Wu et al., 2013*). Both enzymes have one active site per monomer and operate via a two-step mechanism, wherein a linear dinucleotide intermediate is formed, rotated in the active site, then cyclized. Importantly, DncV and cGAS differ in the order in which the nucleotide linkages are formed (*Figure 4A*). DncV initially produces pppA[3′,5′]pG, utilizing ATP as the nucleophile donor and GTP as the electrophile acceptor, whereas cGAS produces pppG[2′,5′]pA (*Gao et al., 2013*; *Kranzusch et al., 2014*). Thus, these two enzymes have opposite preferences for the first phosphodiester bond formed.

In contrast, GacA is a homodimeric enzyme that has one nucleotide substrate binding site per GGDEF domain, or half of the active site per monomer. We observed that GacA generates both types of linear intermediates in the presence of nonhydrolyzable analogs (*Figure 4B*), which means that both ATP and GTP can serve as donor and acceptor. This result reveals a marked difference

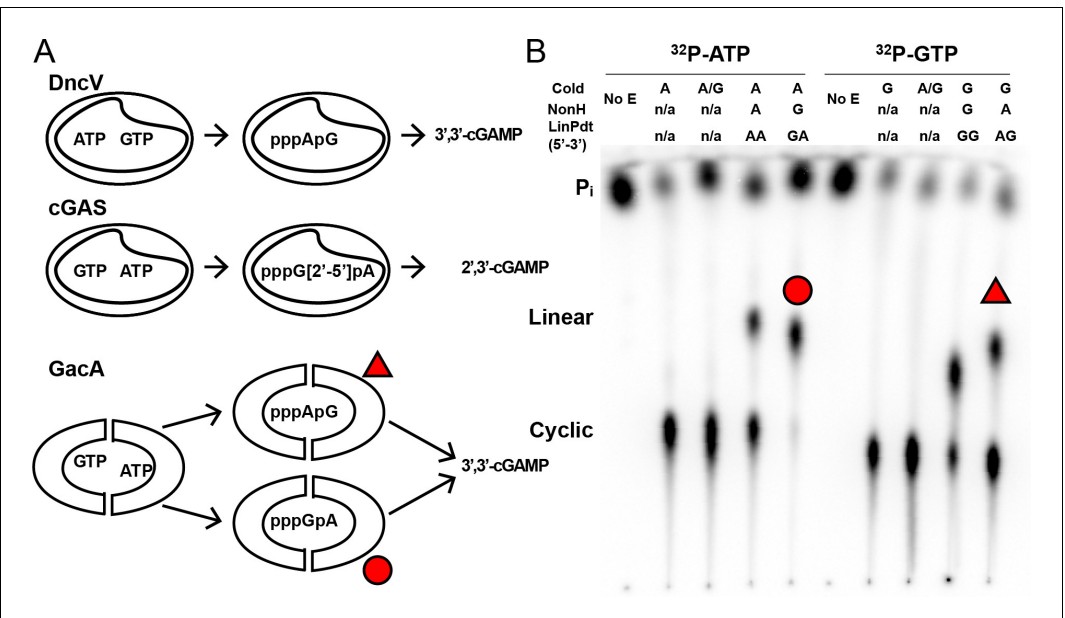

**Figure 4.** Unlike DncV and cGAS, GacA uses either substrate in the first bond-forming step. (A) Reaction pathways to form cyclic GMP-AMP by different dinucleotide cyclases, DncV from *Vibrio cholerae*, cGAS from mammalian cells, and GacA from *Geobacter sulfurreducens*. (B) Cellulose TLC analysis of radiolabeled products from enzymatic reactions with MBP-tagged GacA R393A (I-site mutant) with NTP substrates and nonhydrolyzable analogues. Trace amounts of α-$^{32}$P-labeled ATP or α-$^{32}$P-labeled GTP was doped in the reactions. Reactions were quenched with alkaline phosphatase to digest unreacted nucleotides, resulting in production of inorganic phosphate ($P_i$).

DOI: https://doi.org/10.7554/eLife.43959.008

between GacA and the other two dinucleotide cyclases, DncV and cGAS, and is consistent with the increased promiscuity of GacA to produce homodimeric products, for example cdiG and cdiA.

## Kinetic analysis and mathematical modeling reveal the mechanisms for GacA to produce predominantly cGAMP in vivo

To gain insight into how GacA preferentially produces cGAMP in vivo, it was necessary to establish a full kinetic model for the enzyme (*Figure 5A*). First, we measured initial rates for product formation with single substrates (ATP or GTP) using an enzyme-coupled assay for pyrophosphate detection (*Burns et al., 2014*). In these two cases, the kinetic model is greatly simplified because only one homodimeric product is generated, and this model has been validated for canonical GGDEF enzymes (*Oliveira et al., 2015*). Interestingly, $k_{cat}$ values are similar for production of cdiA and cdiG (0.03–0.04 sec$^{-1}$). The main difference instead appears to be substrate binding, as GTP is the preferred substrate over ATP (*Table 1*). Second, to obtain values for the two heterodimeric equilibrium constants (e.g. $K_{A|G}$, binding constant for ATP given GTP is pre-bound), we compared computationally modeled product ratios to experimental measurements to find $K_{A|G}$ and $K_{G|A}$ values that optimally fit the data (*Figure 5B and C*, *Figure 5—figure supplement 1*). In these models, $k_{cat,AG}$ was set conservatively to equal the catalytic rate constants determined for the homodimeric products (0.03 sec$^{-1}$). This assumption is supported by the fact that ATP and GTP are equally competent as donor and acceptor (*Figure 4B*).

We found that the best-fit values to solve the full kinetic model were $K_{A|G}$ = 71 μM and $K_{G|A}$ = 10 μM (*Table 1*). This result shows that there is positive cooperativity ($K_1 > K_2$) facilitating binding of the second substrate for all reaction pathways. We also analyzed whether binding constants for the second nucleotide are different depending on whether enzyme first bound ATP or GTP, for example comparing $K_{G|A}$ to $K_{2G}$ and $K_{A|G}$ to $K_{2A}$. This effect is termed selective cooperativity, as it affects the product ratios. Comparison of $K_{G|A}$ to $K_{2G}$ shows that there is a 2-fold enhancement of GTP binding to the A-bound vs. G-bound enzyme, which would lead to preferential cGAMP production

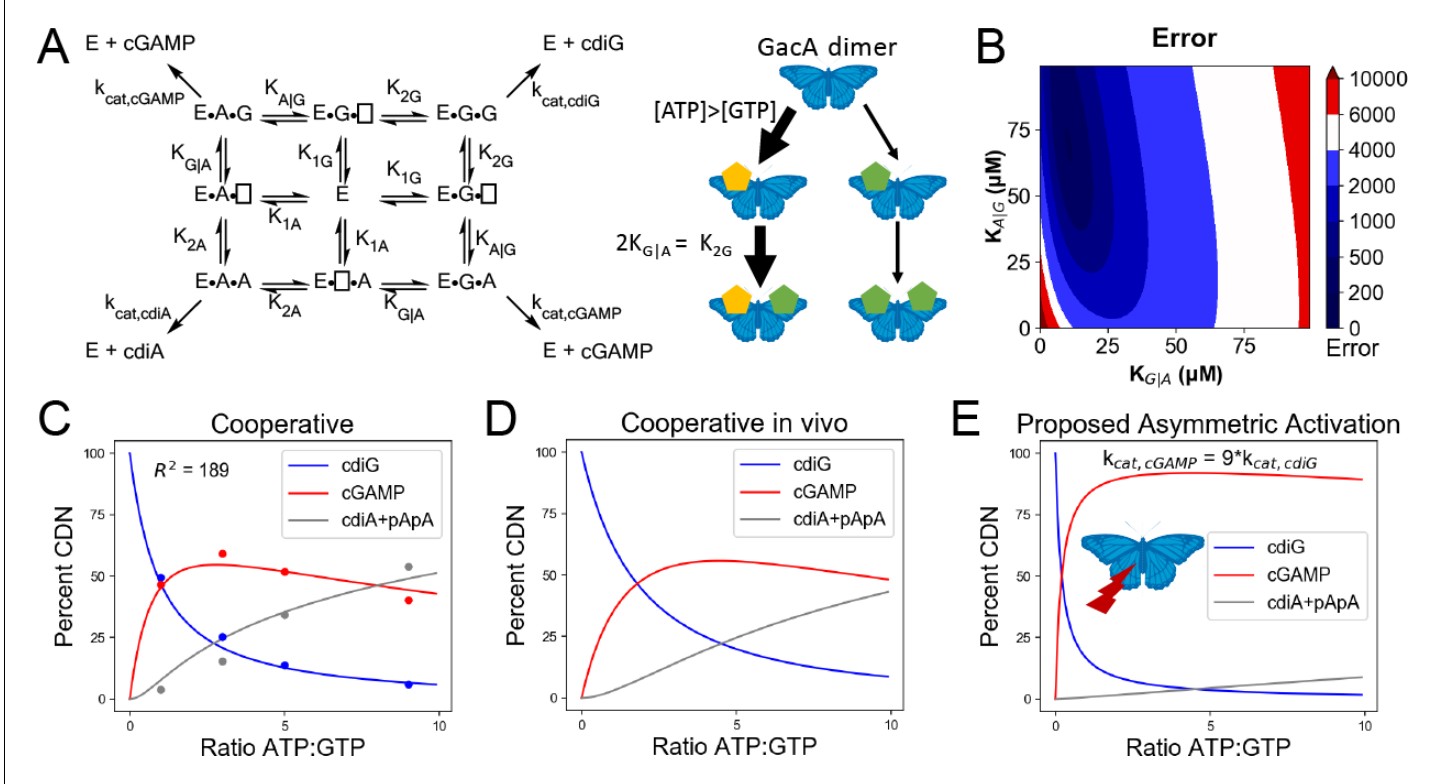

**Figure 5.** Kinetic analysis of GacA reveals that cooperative binding effects lead to preferential cGAMP production. (A) Left, reaction pathways for GacA modeled using Python. 'E' represents the active enzyme, which is a GacA homodimer. 'E•N•[ ]' and 'E•[ ]•N' represent enzyme with NTP bound in the first or second half-active sites, respectively, which are treated as equivalent states. 'E•N•N' represents enzyme with two NTPs bound. The dissociation constant for the first NTP binding event is $K_{1N}$, the second binding event is $K_{2N}$ for homodimeric products, and the second binding event is $K_{X|Y}$ for XTP binding after YTP to produce cGAMP. For example, $K_{G|A}$ is the dissociation constant for GTP given GacA already has ATP bound. Right, schematic summarizing kinetic parameters favoring cGAMP production in vivo. (B) Numerical solution of $K_{G|A}$ and $K_{A|G}$ was obtained by varying them between 0–100 μM and minimizing the least squares error (shown) for the modeled product ratios versus the experimental values. The minimum is observed at $K_{G|A}$ = 10 μM and $K_{A|G}$ = 71 μM, giving the best-fit curve shown in (C). (C) Modeled (lines) versus experimental (points) product ratios for GacA at different ATP to GTP ratios. Data points shown are an average of independent technical replicates (n = 3). (D) Modeled product ratios assuming cellular homeostasis of ATP and GTP levels ($d[ATP]/dt = d[GTP]/dt = 0$). (E) As in (D), except modeled with $k_{cat,AG}$ nine-fold higher than $k_{cat,cdiG}$ or $k_{cat,cdiA}$. With asymmetric activation, GacA could produce cGAMP almost exclusively.

DOI: https://doi.org/10.7554/eLife.43959.009

The following source data and figure supplement are available for figure 5:

**Source data 1.** GacA product ratios.
DOI: https://doi.org/10.7554/eLife.43959.011
**Figure supplement 1.** Kinetic analysis of GacA shows better fit for the cooperative model.
DOI: https://doi.org/10.7554/eLife.43959.010

(*Figure 5A*). Comparison of $K_{A|G}$ to $K_{2A}$ shows a 1.4-fold enhancement of ATP binding to the A-bound vs G-bound enzyme, but in this case the model fit is relatively insensitive to changes in $K_{A|G}$ value (*Figure 5B*), so this effect may or may not be significant. Taken together, the kinetic model provides support for selective cooperativity favoring cGAMP production by enhancing GTP binding to the A-bound enzyme. The A-bound form is favored under cellular conditions where ATP levels are typically 3-fold or more relative to GTP levels (*Buckstein et al., 2008*).

While performing the kinetic modeling, we observed that substrate depletion during in vitro reactions can skew product ratios over time. Since NTP levels are expected to be maintained at cellular homeostasis, we used the computational model to simulate in vivo product ratios with substrate concentrations remaining constant. This model demonstrates that GacA is predominantly a cGAMP synthase across the entire physiological range of substrate ratios (*Figure 5D*), and is unlikely to switch

Biochemistry and Chemical Biology | Microbiology and Infectious Disease

**Table 1.** Kinetic parameters for WT GacA using non-cooperative and cooperative models.

|  | Non-cooperative | Cooperative |
|---|---|---|
| $K_{1A}$, µM | 80 | 343 |
| $K_{2A}$, µM | 80 | 53 |
| $k_{cat,cdiA}$, sec$^{-1}$ | 0.04 | 0.03 |
| $K_{1G}$, µM | 25 | 39 |
| $K_{2G}$, µM | 25 | 20 |
| $k_{cat,cdiG}$, sec$^{-1}$ | 0.04 | 0.03 |
| $K_{A|G}$, µM | 80 | 71 |
| $K_{G|A}$, µM | 25 | 10 |
| $k_{cat,cGAMP}$, sec$^{-1}$ | 0.04 | 0.03 |
| Model RMSD | 4118 | 443 |

DOI: https://doi.org/10.7554/eLife.43959.012

between producing different signals in vivo, as had been an alternative proposed function (*Hallberg et al., 2016*).

The kinetic model also allows us to estimate the impact of receiver (Rec) domain activation. While our data indicate that the traditional phosphorylation site D52 is not required for GacA activity in vivo (*Figure 1D,E*) and in vitro (*Hallberg et al., 2016*), alternative activation of Rec domains by S/T phosphorylation, kinase binding, or ligand binding has been shown (*Lin et al., 2009*; *Ocasio et al., 2015*; *Trajtenberg et al., 2014*; *Wang et al., 2009*). The low $k_{cat}$ values that we measured in vitro are in the range observed for other non-activated GGDEF enzymes (*Wassmann et al., 2007*), and activation has been shown to increase canonical GGDEF activity by up to 50-fold (*Huangyutitham et al., 2013*; *Paul et al., 2007*). While uniform effects on $k_{cat}$ values would not change product ratios, we used the kinetic model to simulate product ratios if $k_{cat,AG}$ was increased asymmetrically by activation more than $k_{cat,diG}$ or $k_{cat,diA}$ (*Figure 5E*). The result is that GacA can be almost fully selective (>90% cGAMP) if the proposed asymmetric activation leads to $k_{cat,AG}$ that is nine times $k_{cat,diG}$ or $k_{cat,diA}$ (*Figure 5E*), which would result from a difference in activation energy ($\Delta\Delta G^{\circ}$) of only 1.3 kcal/mol. Taken together, these results show how cooperative binding, including selective cooperativity induced by the first substrate bound, and tuning substrate affinities to cellular concentrations, could make GacA predominantly produce cGAMP. Mathematical modeling also led to the hypothesis that asymmetric activation could further favor GacA behaving exclusively as a cGAMP signaling enzyme in vivo.

## X-ray crystal structure of Hypr GGDEF domain of *G. metallireducens* GacA bound to guanosine substrate

To gain insight into the molecular basis for function and mechanism of this signaling enzyme, we pursued structural characterization of the GacA GGDEF domain from *Geobacter metallireducens* in the presence of GTP, and obtained a 1.4 Å resolution x-ray crystal structure as an N-terminal fusion with T4 lysozyme (*Table 2*, *Figures 6* and *7*). The GacA GGDEF domain has a βααββαβαβ global topology that positions one guanosine substrate above the signature [G/A/S]G[D/E]E[F/Y] motif and can be overlaid with a canonical GGDEF domain with an RMSD value of 1.152 Å (*Figures 6B* and *8*). A region behind the two alpha helices that support substrate binding is modified from a beta sheet to a helical/loop motif, and varies considerably between GGDEF structures (*Chen et al., 2016*; *Dahlstrom et al., 2015*; *Deepthi et al., 2014*; *Yang et al., 2011*). Electron density for three guanine nucleotides was found in the GacA structure, two in nucleotide-interacting regions that are conserved in other GGDEF domains (*Figure 7*) (*Chan et al., 2004*). One guanine nucleotide is bound near the canonical allosteric inhibitory site (I-site) (*Chan et al., 2004*; *Christen et al., 2006*) and the second nucleotide is bound in the active site above the GGDEF motif (*Figures 6A* and *7*). For the latter, we were only able to find partial localized electron density for the alpha phosphate (*Figure 7B*). It is likely that GTP was hydrolyzed during crystallization but remained coordinated in the active site with density now visible for the guanosine and beta-gamma pyrophosphate (PP$_i$). For

**Table 2.** Crystallographic statistics

| | T4Lys-Gmet_1914 | T4Lys-Gmet_1914 (S Anomalous) |
|---|---|---|
| Data Collection | | |
| Resolution (Å) | 35.75–1.35 (1.37–1.35) | 48.35–2.72 (2.86–2.72) |
| Wavelength (Å) | 1.11582 | 2.25418 |
| Space group | C $2_1$ | C $2_1$ |
| Unit cell dimensions: a, b, c (Å) | 70.68, 111.65, 55.34 | 71.33, 111.96, 55.66 |
| Unit cell dimensions: α, β, γ (°) | 90, 122.75, 90 | 90.0, 123.10, 90.0 |
| Molecules per ASU | 1 | 1 |
| No. reflections: total | 512201 | 61021 |
| No. reflections: unique | 77765 | 9235 |
| Completeness (%) | 97.8 (67.7) | 94.0 (74.9) |
| Multiplicity | 6.6 (3.1) | 6.6 (5.6) |
| $I/\sigma I$ | 21.6 (2.9) | 21.6 (13.8) |
| CC(1/2)[1] (%) | 99.9 (86.2) | 99.4 (99.1) |
| Rpim[2] (%) | 1.7 (22.2) | 2.5 (3.2) |
| No. sites | | 17 |
| Refinement | | |
| Resolution (Å) | 35.10–1.35 | |
| Free reflections (%) | 10 | |
| R-factor/R-free | 15.9/17.3 | |
| R.M.S. deviation: bond distances (Å) | 0.016 | |
| R.M.S. deviation: bond angles (°) | 1.560 | |
| Structure/Stereochemistry | | |
| No. atoms: nonhydrogen, protein | 2783 | |
| No. atoms: ligand | 71 (GTP, GMP, PPi) | |
| No. atoms: water | 440 | |
| Average B-factor: nonhydrogen, protein | 18.7 | |
| Average B-factor: ligand | 28.0 | |
| Average B-factor: water | 33.9 | |
| Ramachandran plot: most favored regions | 97.7% | |
| Ramachandran plot: additionally allowed | 2.3% | |
| Protein Data Bank ID | 5VS9 | |

DOI: https://doi.org/10.7554/eLife.43959.018

clarity, we show both the original and modeled structures with the alpha phosphate. The final guanosine nucleotide binds at the T4 lysozyme-GGDEF interface and may act to stabilize the construct in a way that ATP cannot. To our knowledge, this is the first structure of a Hypr GGDEF domain.

## A Goldilocks model for substrate recognition by GGDEF domains

Our structure of the Hypr GGDEF domain with bound guanosine plus $PP_i$ shows that two specific hydrogen-bonding interactions are made with the nucleobase, a Watson-Crick interaction with Ser348 and a sugar-face interaction with Asn339 (*Figure 6B*). The phosphate backbone is recognized via several specific interactions (*Figure 8*). In particular, a magnesium ion coordinates to the β and γ phosphates and is held in place by the side chain of Asp374 in the GGDEF motif.

A signature difference between Hypr and canonical GGDEF domains is that Hypr GGDEFs have Ser348, whereas the canonical GGDEFs have an aspartate residue at that position. We previously hypothesized that Ser348 would form hydrogen bonds on the Watson-Crick face of either guanine

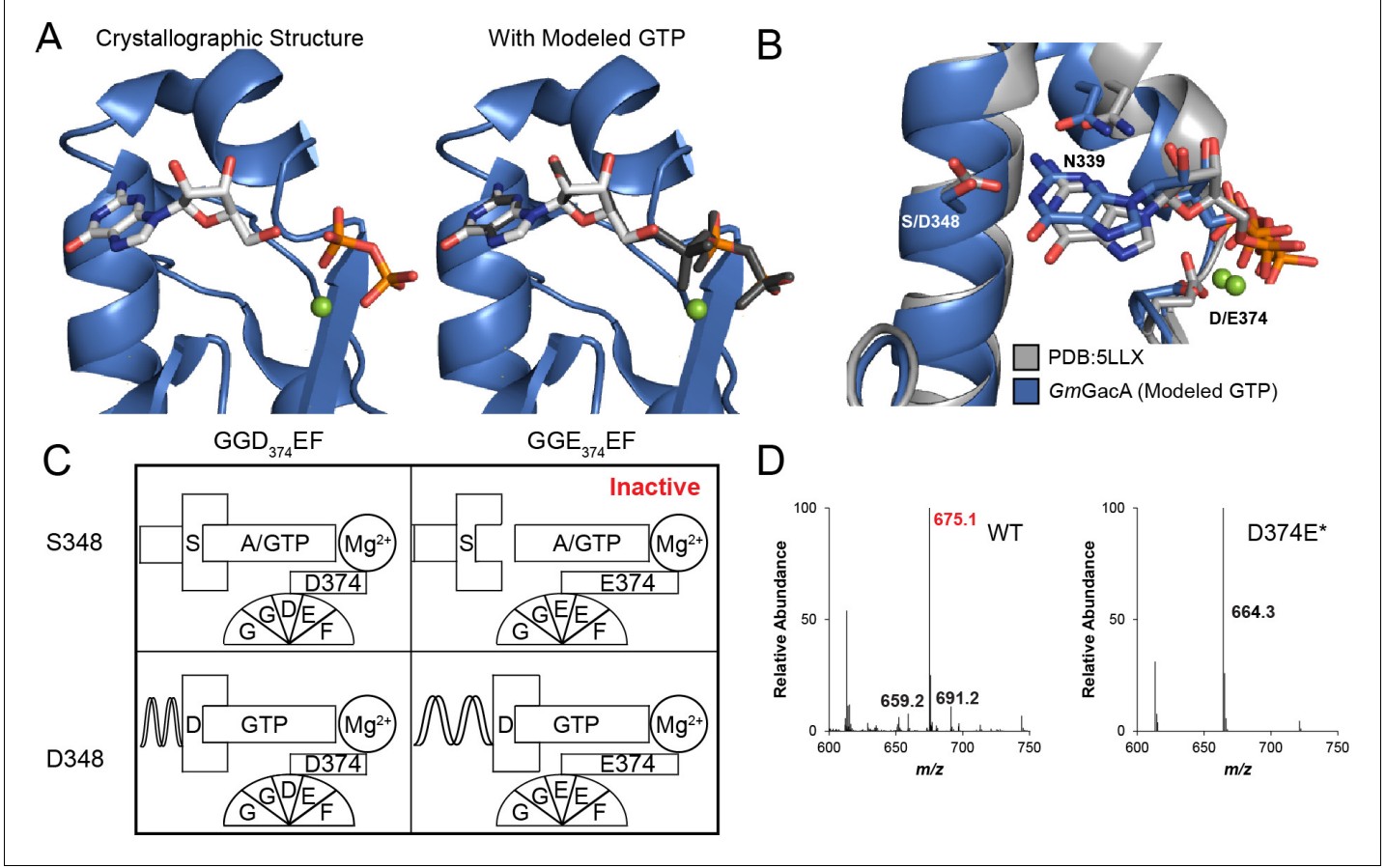

**Figure 6.** The x-ray crystal structure of *Gm*GacA Hypr GGDEF domain support the role of signature residues and the Goldilocks model. (**A**) Active site of the GacA Hypr GGDEF from *G. metallireducens* (Gmet_1914) with guanosine:PPi bound or modeled GTP based on partial alpha phosphate density. (**B**) Superposition of x-ray crystal structures of *G. metallireducens* GacA Hypr GGDEF (blue) and the *Idiomarina* sp. A28L bacteriophytochrome GGDEF (grey, PDB 5LLX), each with bound guanine nucleotides. Key interacting residues are labeled and shown as sticks. (**C**) Schematic illustrating the Goldilocks model that explains why the Ser348/GGEEF combination is inactive. In contrast, the flexible Asp side chain at position 348 permits either GGDEF or GGEEF enzymes to remain active. (**D**) LC/MS analysis of *E. coli* cell extracts overexpressing *G. sulfurreducens* GacA WT or D374E* mutant (* indicates that the numbering used corresponds to the *G. metallireducens* GacA structure, because *Gs*GacA is shorter by one amino acid). Shown are representative MS spectra from integrating the retention time region that would contain the three cyclic dinucleotides. Expected masses are labeled for cdiG (m/z = 691), cGAMP (m/z = 675), and cdiA (m/z = 659). The major peak observed for inactive variants (m/z = 664) is potentially NAD from the lysate.

DOI: https://doi.org/10.7554/eLife.43959.013

The following source data and figure supplement are available for figure 6:

**Source data 1.** LC-MS data for GacA and GGEEF mutant.

DOI: https://doi.org/10.7554/eLife.43959.015

**Figure supplement 1.** In vitro and in vivo analysis of GacA and WspR mutants to test the 'Goldilocks' model.

DOI: https://doi.org/10.7554/eLife.43959.014

or adenine (*Hallberg et al., 2016*), and this can be seen in our structure. In fact, we have shown that Hypr GGDEFs can accept other purine NTPs as substrates (*Figure 8*). However, making the corresponding D-to-S mutant for canonical GGDEFs unexpectedly inactivates the enzyme in four out of five cases (*Hallberg et al., 2016*). One enzyme, GSU3350, remained active, but solely as a diguanylate cyclase.

Overlaying Hypr and canonical GGDEF domains provides a potential explanation for these earlier results (*Figure 6B*), although an important caveat to this analysis is that the Hypr GGDEF structure contains guanosine plus PP$_i$ bound instead of intact GTP in the case of the canonical GGDEF. To interact with serine, which is a shorter side chain than aspartate, the guanosine shifts toward the $\alpha_2$

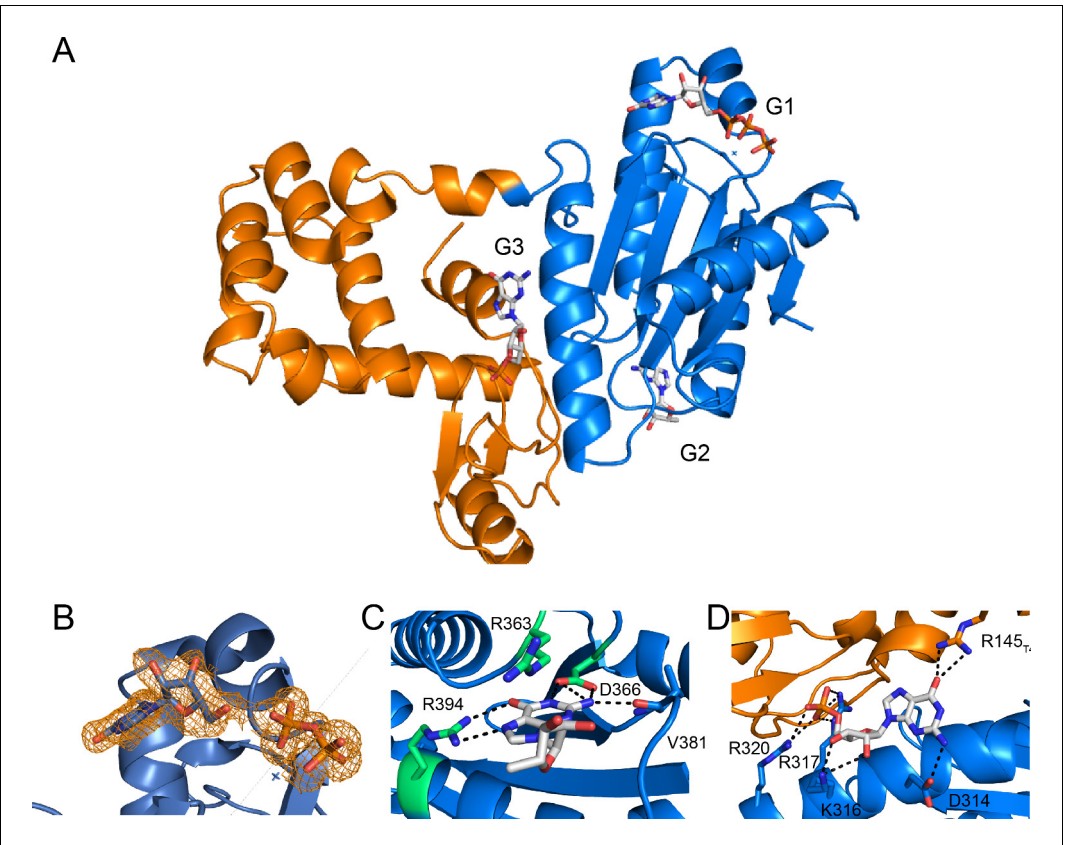

**Figure 7.** Full x-ray crystal structure shows three guanine nucleotides bound. (**A**) Full structure of the T4-Lysozyme-Gmet$_{GGDEF}$ fusion showing the location of three guanine nucleotides (G1: active site; G2: I-site; G3: interface). The T4-Lysozyme is in orange and Gmet$_{GGDEF}$ is in blue. (**B**) GTP binding pocket electron density. An Fo-Fc omit map of electron density contoured at 2.0 σ is shown for the bound G1 nucleotide. (**C**) Interactions of G2 nucleotide bound near the canonical I-site region of the Hypr GGDEF. (**D**) Interactions of G3 nucleotide at the interface between the T4-Lysozyme and GGDEF domains. The interactions between R145$_{T4}$ and D314$_{Gmet1914}$ are expected to occur only for GTP, which may explain our inability to obtain crystals in the presence of ATP.
DOI: https://doi.org/10.7554/eLife.43959.016

helix in the Hypr GGDEF compared to GTP in the canonical structure. This shift maintains the hydrogen-bonding distance (2.6 Å in ImDGC and 2.7 Å in GmGacA), but requires a corresponding shift of other interactions that would stabilize GTP in the active site. In the Hypr GGDEF structure, the Mg$^{2+}$ coordinating the phosphates also moves toward the α$_2$ helix relative to the canonical structure, even though there is no bond between the guanosine and phosphates in our structure.

We hypothesized that this key compensatory shift is due to the presence of Asp374 in the GG<u>D</u>EF motif of GmGacA, which is shorter than the glutamate residue found in the GG<u>E</u>EF motif of the canonical enzyme whose structure was overlaid. To test whether the Mg$^{2+}$-GGDEF interaction is indeed critical to GacA activity, we made the D374*E mutant of GsGacA, which converts the motif to GGEEF (the * indicates that the numbering used corresponds to the G. metallireducens GacA structure, because GsGacA is shorter by one amino acid). The D374*E mutant is inactive, as expected for the Ser348/GG<u>E</u>EF combination causing loss of substrate binding (**Figure 6D**, **Figure 6—figure supplement 1**). The S348*D/D374*E double mutant, which represents the Asp348/GG<u>E</u>EF combination, restores activity, but solely as a diguanylate cyclase. The S348*D mutant also becomes an active diguanylate cyclase.

Taken together, these observations support a 'Goldilocks' model for the GacA active site (**Figure 6C**), in which the residue interacting with the WC face of the nucleotide, Ser348, and the magnesium coordinated the GGDEF motif must be the appropriate distance apart. If the two components are too far apart, as in the Ser348/GG<u>E</u>EF case, substrate binding cannot occur. GacA

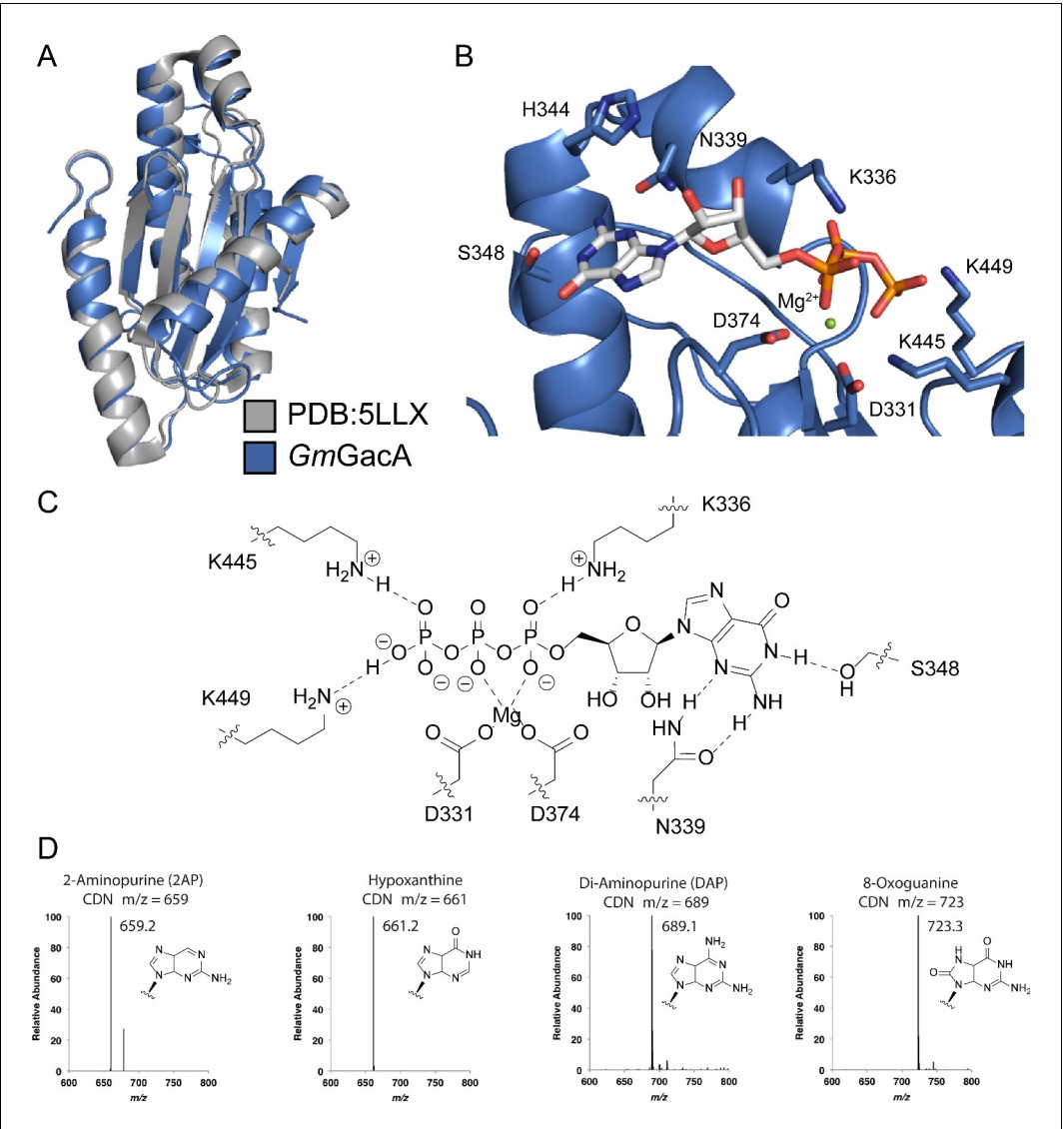

**Figure 8.** Hypr GGDEF domain structure overlay with canonical GGDEF domain; details of the nucleotide binding pocket. (**A**) Overlay of Gmet_1914 GGDEF (blue) with the GGDEF domain of the *I. marina* light-activated GGDEF (Grey, PDB: 5LLX). (**B**) Active site of Gmet_1914 GGDEF in complex with GTP. Residues expected to interact with the GTP or $Mg^{2+}$ cation are labeled. The alpha phosphate electron density was unable to be resolved, but was modeled into the structure. (**C**) Chemical scheme of active site interactions with GTP substrate. (**D**) LC-MS analysis of GacA with various unnatural NTP substrates. The structure of the purine base analog and the mass spectra of the homodimeric CDN produced are shown.

DOI: https://doi.org/10.7554/eLife.43959.017

mutants that represent Asp348/GG<u>E</u>EF and Asp348/GG<u>D</u>EF combinations retain activity because they also can match the right distance, the latter through flexibility of the aspartate side chains. However, these mutants become diguanylate cyclases because the S348D mutation drives specificity for GTP.

In support of this model, bioinformatics analysis shows that natural GGDEF enzymes with Ser/Thr at position 348 harbor the GG<u>DE</u>[F/Y] motif exclusively (*Figure 6—figure supplement 1D*), whereas GGDEF enzymes with Asp at position 348 are almost evenly divided between D and E at the central position of the motif (57% and 43%, respectively) (*Figure 6—figure supplement 1E*). To further demonstrate that the Goldilocks model applies to canonical GGDEF enzymes, we performed mutational analysis on WspR, a diguanylate cyclase from *Pseudomonas aeruginosa* (*Hickman et al.,*

*2005*). The WspR mutants recapitulate the same activity trends that were shown for GacA (*Figure 6—figure supplement 1B*). Furthermore, the model explains our prior D-to-S mutagenesis results; the four inactive enzymes have a GGEEF motif, while the enzyme that retained activity, GSU3350, has a GGDEF motif.

## Cross-dimer interactions affect GGDEF enzyme catalysis and cooperativity

While the Hypr GGDEF monomer structure gave some insights into substrate binding, the enzyme functions as a homodimer, with one NTP binding site per monomer. To elucidate the function of other conserved residues, we superimposed our structure onto both GGDEF domains of the $C_2$ symmetric enzyme dimer structure from *Idiomarina* sp. A28L (*Gourinchas et al., 2017*). In both dimer structures, the glutamate residue that is the fourth residue in GG<u>D</u>EF is close to the GTP/guanosine: PP$_i$ bound to the opposite monomer, and in the case of GacA, is oriented appropriately to deprotonate the 3' hydroxyl group from the substrate (*Figure 9A*). This observation strongly suggests that this glutamate is the general base that activates the nucleophile donor.

This glutamate was among the ultra-conserved residues in GGDEF domains that had no previously assigned function (*Schirmer, 2016*). Mutating this residue to glutamine knocks out catalytic activity of GacA (*Figure 9E*), which is in line with prior experiments demonstrating that this residue is required for canonical GGDEF function (*Malone et al., 2007*). Identifying this glutamate as the general base provides molecular insight into the regulatory mechanisms for GGDEF enzymes. Some GGDEF enzymes are activated by shifting oligomeric states, from monomer to dimer or even to higher order oligomers (*Huangyutitham et al., 2013*; *Paul et al., 2007*). Monomers are inactive because each monomer binds only one NTP. However, other GGDEF enzymes are predicted to be activated by changing the dimer conformation (*Gourinchas et al., 2017*; *Zähringer et al., 2013*). In these cases, the orientation of the two monomers can affect whether the newly identified general base is poised to deprotonate the 3' hydroxyl across the dimer.

We observed another cross-dimer interaction with the guanosine substrate that had different residue identities for Hypr versus canonical GGDEFs. In the diguanylate cyclase dimer, Arg537 appears to form a cation-π interaction by stacking above the nucleobase in the opposite active site (*Figure 9B*). This conserved residue (94% of predicted diguanylate cyclases) also had no prior assigned function (*Schirmer, 2016*). Interestingly, the modeled dimer of the *G. metallireducens* Hypr GGDEF has a tyrosine (Tyr304) at this position, which is tucked away in the monomer structure (*Figure 9C*). However, with side chain rotation it can form a π-π stacking interaction with either adenine or guanine (*Figure 9D*, *Figure 9—figure supplement 1*). Thus, our analysis of the structures suggests that Arg537 is a previously unappreciated determinant of substrate specificity in diguanylate cyclases, which is replaced by other residues in Hypr GGDEF enzymes.

In support of this functional assignment, the corresponding Y304*R mutant was found to have a product ratio more skewed towards cdiG, which is consistent with the cation-π interaction favoring guanine over adenine (*Figure 9E*, *Figure 9—figure supplement 1C*). Also, an analysis of Hypr GGDEFs from different bacteria previously showed that two enzymes harboring an arginine produce more cdiG (Cabther_A1065 and Ddes1475), whereas enzymes harboring tyrosine, serine, alanine, or glutamine produce predominantly cGAMP (*Hallberg et al., 2016*). These results reveal that this cross-dimer interaction affects product distribution, leading us to propose a mechanism for the cooperative binding and putative asymmetric activation effects shown by kinetic modeling. As shown in the model, the status of one monomer, for example the identity of nucleotide substrate and/or Rec activation, can be communicated by residue(s) that make cross-dimer interactions to the substrate in the other monomer's active site. For example, changes to the orientation of Tyr304 will tune the binding energy, possibly in a differential manner for guanosine or adenosine substrates. There may be other residues besides Tyr304 that are involved in this cross-dimer communication. Taken together, our analysis of the crystal structure shows that consideration of cross-dimer interactions may be key to unlocking residue functions for both Hypr and canonical GGDEFs.

## Conclusions

For bacteria, obtaining energy is key to niche survival, whether in a host or outside environment. In dynamic anaerobic environments, where oxygen is unavailable or limiting, microbes must seek

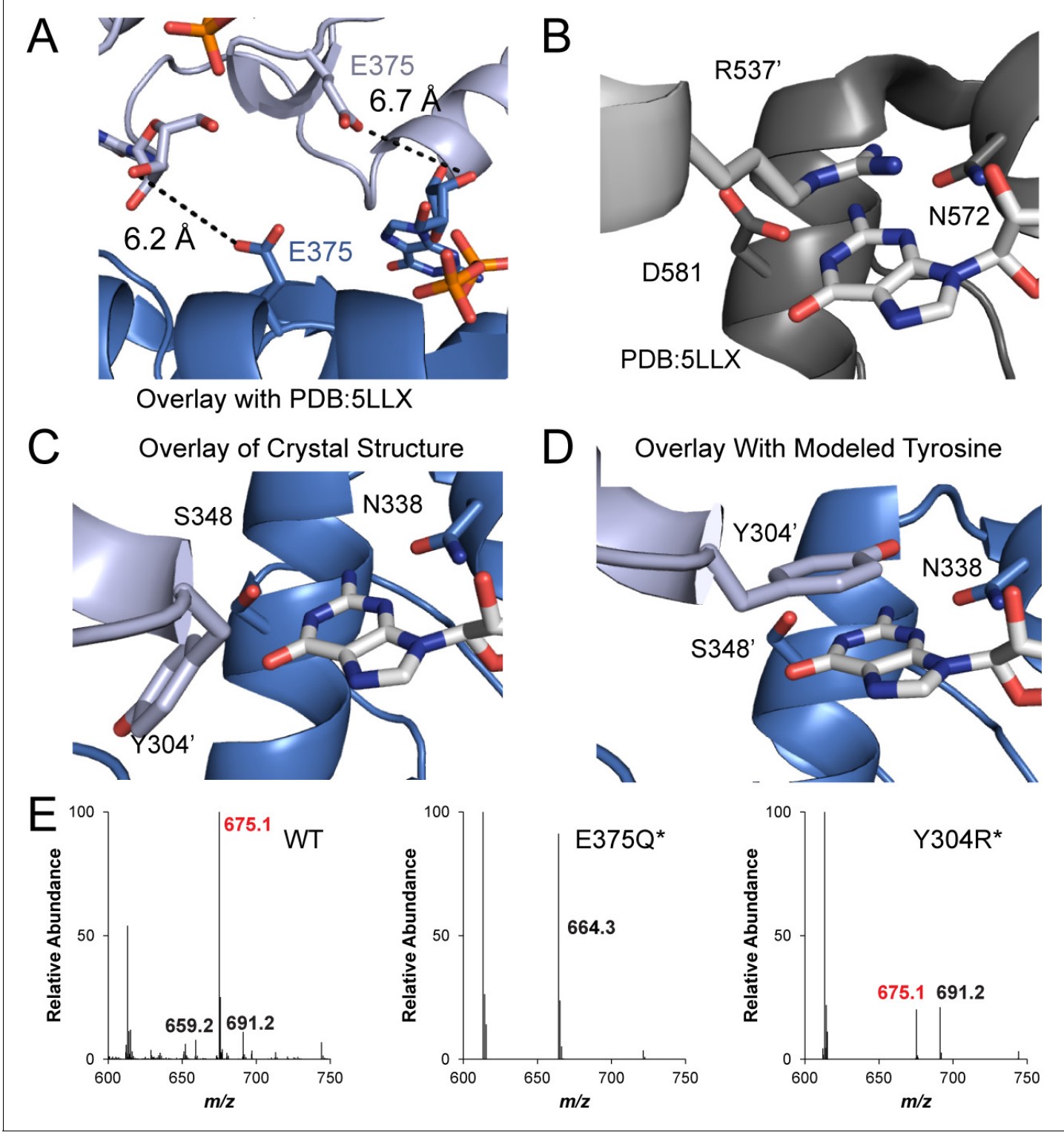

**Figure 9.** The *Gm*GacA Hypr GGDEF domain structure reveals the molecular basis for hydroxyl activation and substrate-assisted cooperativity for GGDEF enzymes via cross-dimer interactions. (**A**) The modeled dimer of GacA GGDEF domains shows that the Glu375 residue from one monomer is poised to deprotonate the 3'-OH of GTP bound in the other half active site. The blue GTP is bound by the blue protein monomer, whereas the grey-blue GTP is bound by the grey-blue protein monomer. (**B**) and (**C**) Comparison of the crystallized GGDEF dimer (grey/dark grey, PDB 5LLX) and the modeled GacA dimer (blue/grey-blue) reveals another cross-dimer residue in the half active site that has a different identity between the canonical (D581) and Hypr (Y304') GGDEFs. (**D**) Y304' was rotated 180 degrees about the alpha carbon bond from the crystallized structure. (**E**) LC/MS analysis of
*Figure 9 continued on next page*

*Figure 9 continued*

*E. coli* cell extracts overexpressing *G. sulfurreducens* GacA WT or cross-dimer mutants (* indicates that the numbering used corresponds to the *G. metallireducens* GacA structure, because *Gs*GacA is shorter by one amino acid). Shown are representative MS spectra from integrating the retention time region that would contain the three cyclic dinucleotides. Expected masses are labeled for cdiG (m/z = 691), cGAMP (m/z = 675), and cdiA (m/z = 659). The major peak observed for inactive variants (m/z = 664) is potentially NAD from the lysate.

DOI: https://doi.org/10.7554/eLife.43959.019

The following source data and figure supplement are available for figure 9:

**Source data 1.** LC-MS data for GacA and cross-dimer mutants.
DOI: https://doi.org/10.7554/eLife.43959.021
**Figure supplement 1.** Identification and analysis of a critical cross-dimer interaction with substrate in the active site.
DOI: https://doi.org/10.7554/eLife.43959.020

alternative electron acceptors. One such strategy that profoundly impacts Earth's biogeochemistry is the process of extracellular electron transfer to metals, surfaces, and other cells. Use of environmental metal oxides as terminal electron acceptors by *Geobacter* requires cell-metal contact to facilitate electron transfer, and while attachment to surfaces is typically regulated by cdiG signaling, our results demonstrate that a separate mechanism has emerged for metal particle attachment. In retrospect, permanent biofilm-like attachment as driven by cdiG signaling would be a poor choice for interacting with environmental metal oxides, as Fe(III) oxides are usually nanophase (<100 nm), and a single metal particle cannot provide enough energy to support cell division (*Levar, 2013*; *Zacharoff et al., 2017*). Thus, based on energetics and size, metal oxides present a conundrum for metal-reducing bacteria: a surface that requires transient, rather than permanent, contact.

We hypothesize that cGAMP signaling — and thus GacA — arose as a divergent signaling system for this separate, transiently surface-associated state (*Figure 10*). Specifically, GacA helps coordinate electron transfer to Fe(III) oxides, but is not involved in permanent biofilm growth on electrodes, or planktonic growth with soluble metals. These phenotypes contrast with the involvement of a canonical cdiG-synthesizing GGDEF enzyme, EsnD, in biofilm-based electricity production on electrodes (*Chan et al., 2017*). Along with growing evidence that transient-attached and permanent-attached states are distinct stages in the biofilm lifestyles of bacteria (*Lee et al., 2018*), this study provides the first evidence that these modes can be signaled by two different cyclic dinucleotides. Whether this paradigm is more widespread or whether different mechanisms are present in other bacteria is the subject of future work.

Interestingly, *Vibrio cholerae* has a completely distinct cGAMP signaling pathway from the GacA-cGAMP-riboswitch pathway analyzed in this study (*Davies et al., 2012*; *Severin et al., 2018*). As exemplified by GacA, Hypr GGDEFs likely arose from divergent evolution of diguanylate cyclase enzymes in ancestral deltaproteobacteria, as it is conserved across species of *Geobacter*, *Myxobacteria*, and others. In contrast, DncV is found in a pathogenicity island unique to the El Tor strain of *V. cholerae* that also contains a cGAMP-activated phospholipase (*Severin et al., 2018*). Activation of this phospholipase changes membrane fatty acid composition and inhibits cell growth (*Severin et al., 2018*), which contrasts with the riboswitch-driven transcriptional response and electron transfer phenotype in *G. sulfurreducens*.

A challenge we faced at the outset was to reconcile our in vivo observations that showed GacA produces cGAMP-specific phenotypes with prior in vitro observations that showed GacA to be a promiscuous dinucleotide cyclase. This paradox mirrors a common problem in studying two-component signaling: histidine kinases can phosphorylate non-cognate receiver domains in vitro, whereas this crosstalk is not observed in vivo. By combining biochemical analysis with mathematical modeling, we demonstrate that under standard cellular conditions, substrate-assisted cooperative binding biases production to give predominantly cGAMP in vivo (*Figure 5A*). The main side product, cdiG, is likely produced by GacA at sufficiently low amounts that a housekeeping phosphodiesterase can prevent cross-signaling in vivo, as shown for PdeH in *E. coli* (*Sarenko et al., 2017*). A new hypothesis that arises from the model is that with asymmetric activation, GacA can act as a completely selective enzyme in vivo (*Figure 5E*). One molecular mechanism that we propose here and would be very intriguing to explore in the future is that asymmetric activation could occur via modulation of a single Rec domain. These results run counter to the intuition that enzymes with homodimeric active sites can only produce symmetrical products. In fact, they lead to a newly intuitive explanation: active site

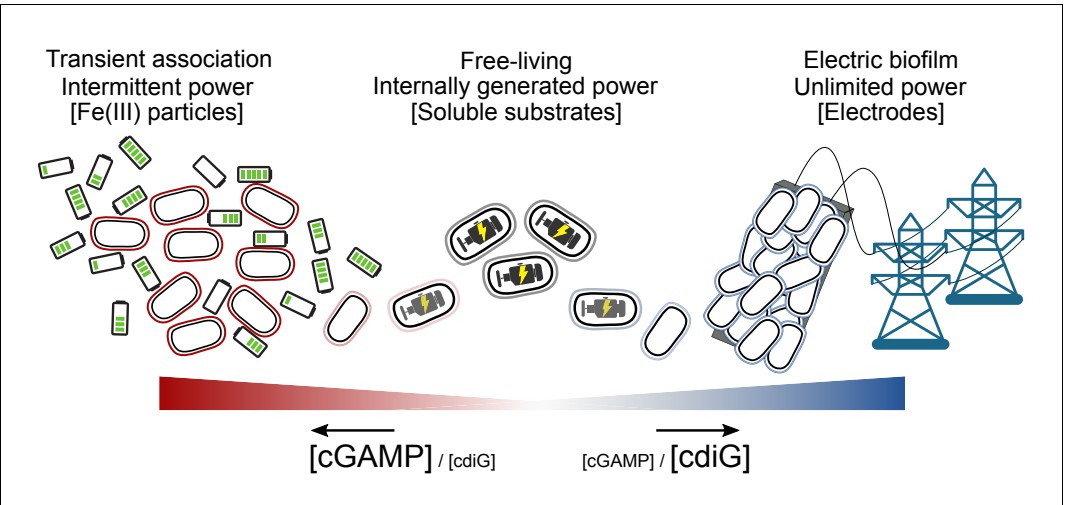

**Figure 10.** Proposed transient energy lifestyle controlled by cyclic GMP-AMP signaling that is distinct from cyclic di-GMP controlled electrical biofilm. The second messenger cGAMP is synthesized by GacA, a Hypr GGDEF enzyme, and enhances Fe(III) particle reduction but is not required for growth on electrodes. Cyclic GMP-AMP enhances the transcription of over 30 genes by binding to riboswitches upstream of these genes, including cytochromes and pili. Cyclic di-GMP, commonly used by bacteria for permanently attached lifestyles, is important for growth on electrodes where there is an infinite supply of electron acceptor, but is not required for Fe(III) particle reduction where each particle has a finite electron accepting capacity. To our knowledge, this is the first time that these two contact-dependent electron transfer processes have been shown to be differentially regulated on a global scale.

DOI: https://doi.org/10.7554/eLife.43959.022

symmetry is broken once the first substrate binds, and the identity of that substrate can influence the second binding event, giving rise to substrate-assisted selectivity. Our structural analysis further reveals a signature cross-dimer residue in Hypr GGDEFs that is poised to 'read' substrate identity and allosterically transfer that information to the other half active site (*Figure 9B–D*).

Combining structural and biochemical analyses also led to several insights into the function of GGDEF enzymes in general, in terms of substrate recognition and catalysis. The Goldilocks model explains why Ser348/GGEEF enzymes are non-functional and provides a basis of selectivity against pyrimidine substrates. The natural pyrimidine NTPs most likely are too short to interact with both the active site $Mg^{2+}$ and residues that recognize the nucleobase, even if hydrogen bonds were matched. GGDEF enzymes capable of coordinating larger divalent metals or otherwise shortening the distance may be able to accommodate pyrimidine substrates. The identification of cross-dimer residues as the general base and involved in substrate recognition provides a molecular basis for activation mechanisms that involve conformational changes of the dimer.

From the broader perspectives of protein evolution and engineering, GacA provides an important case study for divergent evolution to a new in vivo function. Our results reveal functional intermediates in a potential mutational pathway to evolve cGAMP cyclases from diguanylate cyclases. First, only GGDEF not GGEEF enzymes are on pathway, followed by R304Y and D348S in either order. Importantly, this implies that cGAMP signaling can arise in any bacteria harboring GGDEF-type diguanylate cyclases. Another major finding is that this 'promiscuous' enzyme is the functional endpoint and in fact is attuned to play a highly specific role in the cell, as shown by phenotypic data that support its key role in cGAMP production and signaling. However, in vitro analysis or biochemical screening that does not account for cellular substrate concentration and homeostasis would result in an incorrect functional assignment. In fact, this in vitro 'blindspot' caused Hypr GGDEF enzymes to remain undiscovered until we performed an in vivo biosensor-based screen (*Hallberg et al., 2016*). Taken together, these insights are instructive for future efforts to discover and design signaling enzymes that produce other cyclic dinucleotides, besides the four currently known in bacteria and mammals.

# Materials and methods

## Key resources table

| Reagent type (species) or resource | Designation | Source or reference | Additional information |
|---|---|---|---|
| Strain, strain background (*G. sulfurreducens*) | | Laboratory collection | Cell line maintained in D. bond lab |
| Strain, strain background (*G. sulfurreducens*) | Tn7::cGAMP-nanoluc | this work | Integrated using pCGAMP-9 |
| Strain, strain background (*G. sulfurreducens*) | Tn7::cdiG-nanoluc | this work | Integrated using pGGv2-2 |
| Strain, strain background (*G. sulfurreducens*) | Δ*gacA* | this work | Deleted using pDGSU1658 |
| Strain, strain background (*G. sulfurreducens*) | Δ*gacA* Tn7:: *gacA*[OE] | this work | Integrated using pGSU1658-5 |
| Strain, strain background (*G. sulfurreducens*) | Δ*gacA* Tn7::*gacA*[WT] | this work | Integrated using pGSU1658-6 |
| Strain, strain background (*G. sulfurreducens*) | Δ*gacA* Tn7::*gacA*[D52A] | this work | Integrated using pGSU1658-7 |
| Strain, strain background (*G. sulfurreducens*) | Δ*gacA* Tn7::*gacA*[R393A] | this work | Integrated using pGSU1658-13 |
| Strain, strain background (*G. sulfurreducens*) | Δ*gacA* Tn7::cAG-nanoluc | this work | Integrated using pCGAMP-9 |
| Strain, strain background (*G. sulfurreducens*) | Δ*gacA* Tn7::cdiG-nanoluc | this work | Integrated using pGGv2-2 |
| Strain, strain background (*G. sulfurreducens*) | Δ*gacA* Tn7::cAG-nanoluc / pGacA[+] | this work | |
| Strain, strain background (*G. sulfurreducens*) | Δ*gacA* Tn7::cAG-nanoluc/pGacA[WT] | this work | |
| Strain, strain background (*G. sulfurreducens*) | Δ*gacA* Tn7::cAG-nanoluc / pGacA[D52A] | this work | |
| Strain, strain background (*G. sulfurreducens*) | Δ*gacA* Tn7::cAG-nanoluc / pGacA[R393A] | this work | |
| Strain, strain background (*G. sulfurreducens*) | Δ*esnD* | *Chan et al. (2017)* | |
| Strain, strain background (*G. sulfurreducens*) | Δ*esnD* Tn7::*esnD*[OE] | this work | Integrated using pGSU3376-4 |
| Strain, strain background (*G. sulfurreducens*) | Δ*esnD* Tn7::cAG-nanoluc | this work | Integrated using pCGAMP-9 |
| Strain, strain background (*G. sulfurreducens*) | Δ*esnD* Tn7::cdiG-nanoluc | this work | Integrated using pGGv2-2 |

*Continued on next page*

*Continued*

| Reagent type (species) or resource | Designation | Source or reference | Additional information |
|---|---|---|---|
| Strain, strain background (*E. coli*) | S17-1; *recA pro hsdR RP4-2-Tc*::Mu-Km::Tn7 | *Simon et al. (1983)* | Donor strain |
| Strain, strain background (*E. coli*) | MFDpir; RP4-2-Tc::[Δ*Mu1*::*aac(3)IV*-Δ*aphA*-Δ*nic35*-Δ*Mu2*::*zeo*] Δ*dapA*::(*erm-pir*) Δ*recA* | *Ferrières et al. (2010)* | Donor strain for Tn7 integration |
| Strain, strain background (*E. coli*) | MFDpir/pTNS3; MFD*pir* with plasmid expressing *tnsABCD* | *Choi et al. (2008)* | Used for integration downstream of *glmS* |
| Strain, strain background (*E. coli*) | BL21(DE3) star | Life Technologies | |
| Strain, strain background (*E. coli*) | BL21(DE3) star pET31b-Dp17; pCOLADuet-GSU1658 | *Hallberg et al. (2016)* | For flow cytometry analysis |
| Strain, strain background (*E. coli*) | BL21(DE3) star pET31b-Gm790p1-4delA; pCOLADuet-GSU1658 | *Hallberg et al. (2016)* | For flow cytometry analysis |
| Strain, strain background (*E. coli*) | BL21(DE3) star pET31b-Dp17; pCOLADuet-WspR | this work | For flow cytometry analysis |
| Strain, strain background (*E. coli*) | BL21(DE3) star pET31b-Gm790p1-4delA; pCOLADuet-WspR | this work | For flow cytometry analysis |
| Strain, strain background (*E. coli*) | BL21(DE3) star pET31b-Dp17; pCOLADuet-WspR D226S | this work | For flow cytometry analysis |
| Strain, strain background (*E. coli*) | BL21(DE3) star pET31b-Gm790p1-4delA; pCOLADuet-WspR D226S | this work | For flow cytometry analysis |
| Strain, strain background (*E. coli*) | BL21(DE3) star pET31b-Dp17; pCOLADuet-WspR E370D | this work | For flow cytometry analysis |
| Strain, strain background (*E. coli*) | BL21(DE3) star pET31b-Gm790p1-4delA; pCOLADuet-WspR E370D | this work | For flow cytometry analysis |
| Strain, strain background (*E. coli*) | BL21(DE3) star pET31b-Dp17; pCOLADuet-WspR D226S/E370D | this work | For flow cytometry analysis |
| Strain, strain background (*E. coli*) | BL21(DE3) star pET31b-Gm790p1-4delA; pCOLADuet-WspR D226S/E370D | this work | For flow cytometry analysis |
| Recombinant DNA reagent | pRK2-Geo2 | | Geobacter expression vector |
| Recombinant DNA reagent | pTn7C146 | | Tn7 integrative vector, derivative of pTJ1 |
| Recombinant DNA reagent | pGSU1658-1 (pGacA+) | | GSU1658 (*gacA*) in pRK2-Geo2 under the control of the *acpP* promoter |
| Recombinant DNA reagent | pGSU1658-8 (pGacA^WT) | | GSU1658 (*gacA*) in pRK2-Geo2 under the control of the native *gacA* promoter |

*Continued on next page*

Continued

| Reagent type (species) or resource | Designation | Source or reference | Additional information |
|---|---|---|---|
| Recombinant DNA reagent | pGSU1658-9 (pGacA$^{D52A}$) | | GSU1658 ($gacA^{D52A}$) in pRK2-Geo2 under the control of the native $gacA$ promoter |
| Recombinant DNA reagent | pGSU1658-17 (pGacA$^{R393A}$) | | GSU1658 ($gacA^{R393A}$) in pRK2-Geo2 under the control of the native $gacA$ promoter |
| Recombinant DNA reagent | pGSU3376-1 | | GSU3376 ($esnD^{OE}$) in pRK2-Geo2 under the control of the $acpP$ promoter |
| Recombinant DNA reagent | pGSU1658-5 | this work | GSU1658 ($gacA^{OE}$) under the control of the $acpP$ promoter in Tn7 integrative vector |
| Recombinant DNA reagent | pGSU1658-6 | this work | GSU1658 ($gacA$) under the control of the native $gacA$ promoter in Tn7 integrative vector |
| Recombinant DNA reagent | pGSU1658-7 | this work | GSU1658 ($gacA^{D52A}$) under the control of the native $gacA$ promoter in Tn7 integrative vector |
| Recombinant DNA reagent | pGSU1658-13 | this work | GSU1658 ($gacA^{R393A}$) under the control of the native $gacA$ promoter in Tn7 integrative vector |
| Recombinant DNA reagent | pGSU3376-4 | this work | GSU3376 ($esnD^{OE}$) under the control of the $acpP$ promoter in Tn7 integrative vector |
| Recombinant DNA reagent | pK18mobsacB | *Simon et al. (1983)* | $sacB$ suicide vector for gene deletion |
| Recombinant DNA reagent | pDGSU1658 | this work | Flanking regions of GSU1658 in pK18mobsacB |
| Recombinant DNA reagent | pET-MBP-GSU1658 R393A | *Hallberg et al. (2016)* | Modified pET16a vector containing the GSU1658 R393A mutant with an N-terminal 6xHis-MBP tag under the control of the T7 promoter |
| Recombinant DNA reagent | pET-MBP-GSU1658 S347D/R393A | this work | Modified pET16a vector containing the GSU1658 S347D/R393A mutant with an N-terminal 6xHis-MBP tag under the control of the T7 promoter |
| Recombinant DNA reagent | pET-MBP-GSU1658 D373E/R393A | this work | Modified pET16a vector containing the GSU1658 D373E/R393A mutant with an N-terminal 6xHis-MBP tag under the control of the T7 promoter |
| Recombinant DNA reagent | pET-MBP-GSU1658 S347D/D373E/R393A | this work | Modified pET16a vector containing the GSU1658 S347D/D373E/R393A mutant with an N-terminal 6xHis-MBP tag under the control of the T7 promoter |

*Continued on next page*

Continued

| Reagent type (species) or resource | Designation | Source or reference | Additional information |
|---|---|---|---|
| Recombinant DNA reagent | pET24a T4Lysozyme-GmetGGDEF | this work | pET24a vector containing the coding sequence for a chimeric protein consisting an N-terminal T4 lysozyme E11Q mutant followed by residues 294–459 of Gmet_1914 |
| Recombinant DNA reagent | pET24a GSU1658 | *Hallberg et al. (2016)* | pET24a vector containing the WT GSU1658 coding sequence with a C-terminal 6xHis tag under the control of the T7 promoter |
| Recombinant DNA reagent | pET24a GSU1658 D373E | this work | pET24a vector containing the GSU1658 D373E coding sequence with a C-terminal 6xHis tag under the control of the T7 promoter |
| Recombinant DNA reagent | pET24a GSU1658 E374Q | this work | pET24a vector containing the GSU1658 E374Q coding sequence with a C-terminal 6xHis tag under the control of the T7 promoter |
| Recombinant DNA reagent | pET24a GSU1658 Y303R | this work | pET24a vector containing the Y303R GSU1658 coding sequence with a C-terminal 6xHis tag under the control of the T7 promoter |
| Recombinant DNA reagent | pCOLADuet-1 GSU1658 | *Hallberg et al. (2016)* | pCOLADuet-1 vector containing the WT GSU1658 coding sequence between the NdeI and XhoI restriction sites. |
| Recombinant DNA reagent | pCOLADuet-1 WspR | this work | pCOLADuet-1 vector containing the codon-optimized WT WspR coding sequence between the NdeI and XhoI restriction sites. |
| Recombinant DNA reagent | pCOLADuet-1 WspR D226S | this work | pCOLADuet-1 vector containing the codon-optimized D226S WspR coding sequence between the NdeI and XhoI restriction sites. |
| Recombinant DNA reagent | pCOLADuet-1 WspR E370D | this work | pCOLADuet-1 vector containing the codon-optimized E370 WspR coding sequence between the NdeI and XhoI restriction sites. |
| Recombinant DNA reagent | pCOLADuet-1 WspR D226S/E370D | this work | pCOLADuet-1 vector containing the codon-optimized D226S/E370D WspR coding sequence between the NdeI and XhoI restriction sites. |
| Recombinant DNA reagent | (pCGAMP-1) | this work | The promoter of GSU1761 with cAG selective GEMM-1b riboswitch cloned upstream of nanoluciferase in pTOPO2.1 |
| Recombinant DNA reagent | (pCGAMP-9) | this work | cAG reporter-nanoluc fusion in pTn7C146, subcloned from pCAG-1 |

*Continued on next page*

*Continued*

| Reagent type (species) or resource | Designation | Source or reference | Additional information |
|---|---|---|---|
| Recombinant DNA reagent | pGGv2-1 | this work | A cdiG selective variant (A20G) GEMM-1b of Gmet_0970 replaced the GSU1761 GEMM-1b riboswitch cloned upstream of nanoluciferase in pTOPO2.1 |
| Recombinant DNA reagent | pGGv2-2 | this work | cdiG reporter-nanoluc fusion in pTn7C146 subcloned from pGGv2-1 |
| Recombinant DNA reagent | pET31b-Gm790p1-4delA | *Kellenberger et al. (2015)* | pET31b vector expressing the Spinach1-GM790p1-4delA (cAG-selective) biosensor |
| Recombinant DNA reagent | pET31b-Dp17 | *Wang et al. (2016)* | pET31b vector expressing the Spinach2-Dp17 (cdiG-selective) biosensor |
| Sequence-based reagent | Codon-optimized WspR (oligonucleotide) | IDT | ATGCATAATCCGCATGAATCAAA GACGGACCTGGGAGCTCCACTT GACGGAGCCGTGATGGTTTTATT AGTGGACGACCAGGCGATGATCG GTGAGGCGGTCCGCCGTTCTCTG GCTTCTGAAGCGGGCATCGACTTC CATTTTTGCTCCGATCCGCAGCAA GCGGTAGCGGTAGCCAATCAAATT AAGCCCACGGTTATCCTGCAGGAT CTGGTCATGCCTGGCGTGGATGG GCTGACATTGTTAGCAGCTTATCG CGGAAACCCTGCAACACGCGACAT TCCGATCATTGTGCTGAGTACCAA GGAGGAACCCACTGTTAAGTCAGC TGCATTTGCAGCCGGGGCGAATG TGCATTTGCAGCCGGGGCGAATG ACTACCTGGTCAAACTTCCAGATG CGATCGAATTAGTTGCTCGCATCC GCTACCACAGTCGCAGCTACATCG CGCTTCAGCAACGCGATGAAGCCT ACCGCGCCTTGCGCGAATCCCAGC AGCAGCTTCTTGAAACGAACCTGG TTTTGCAGCGTCTGATGAACTCCG ACGGTTTAACGGGTTTGTCTAATC GCCGTCATTTTGATGAATACTTAG AGATGGAATGGCGTCGTAGTTTGC GTGAACAATCTCAGTTGTCATTACT TATGATCGACGTCGACTACTTTAAA TCGTACAACGATACCTTCGGCCATG TAGCGGGTGACGAAGCATTACGTC AAGTCGCTGGCGCGATCCGTGAAGG GTGCTCCCGTTCTTCTGACCTTGCG GCTCGCTATGGTGGAGAGGAGTTTG CAATGGTTCTGCCTGGGACATCACCG GGGGGCGCTCGCCTGTTGGCTGAGA AAGTGCGTCGCACGGTGGAAAGTTTG CAGATCTCGCATGATCAACCGCGTCCA GGCTCGCATTTAACGGTGTCGATCGGC GTATCCACCTTGGTTCCTGGAGGTGGA GGCCAGACCTTTCGCGTTTTGATCGAA ATGGCTGACCAGGCATTATACCAGGCC AAAAATAATGGACGTAATCAGGTGGGA TTGATGGAACAACCAGTACCTCCGGCA CCTGCTGGA |

## General reagents and oligonucleotides

All oligonucleotides were purchased from Elim Biopharmaceuticals (Hayward, CA) or IDT (Coralville, IA). The codon-optimized WspR gene was purchased from IDT as a gBlock (See Key Resources Table). Cyclic dinucleotide standards were purchased from Axxora (Farmingdale, NY) or enzymatically synthesized. NTP stocks were purchased from New England Biolabs (Boston, MA).

## Growth and medium conditions

All strains and plasmids used in this study are listed in the Key Resources Table. Antibiotics were used in the following concentration for *E. coli*; kanamycin 50 µg/mL; spectinomycin 50 µg/mL, chloramphenicol 25 µg/mL, carbenicillin 50 µg/mL, and ampicillin 100 µg/mL. For *G. sulfurreducens*; kanamycin 200 µg/mL and spectinomycin 50 µg/mL. *G. sulfurreducens* strains and mutants were grown in anoxic medium with excess acetate (20 mM) and limiting fumarate (40 mM) as described (*Chan et al., 2015*). Agar (1.5%) was added to the acetate-fumarate medium to culture for clonal isolates on semisolid surface in a $H_2:CO_2:N_2$ (5:20:75) atmosphere in an anaerobic workstation (Don Whitley). All growth analyses were initiated by picking a single colony from acetate-fumarate agar using freshly streaked, −80°C culture stocks. When electrodes were used as the electron acceptor, fumarate was replaced with 50 mM NaCl to maintain a similar ionic strength.

When insoluble Fe(III) oxide or soluble Fe(III) citrate was used as the electron acceptor, a non-chelated mineral mix was used (*Chan et al., 2015*). XRD amorphous insoluble Fe(III)-(oxyhydr)oxide was produced by first synthesizing Schwertmannite ($Fe_8O_8(OH)_6(SO_4)\cdot nH_2O$), combining 10 g of Fe(II) sulfate in 1 L of water with 5.5 mL of 30% $H_2O_2$ overnight (*Levar et al., 2017*). The solids were centrifuged at $3,700 \times g$ and re-suspended in $dH_2O$ three times to obtain Schwertmannite in a pH ~5 solution. This stable product could be added to basal medium and sterilized by autoclaving, resulting in ~30 mM Fe(III) (based on Fe(III) extractable by $NH_3OH$). Autoclaving in pH 7 basal medium converts Schwertmannite into a high surface area, XRD amorphous insoluble Fe(III)-(oxyhydr)oxide and is the primary form of insoluble Fe(III) oxides presented in this study, which we refer to as simply Fe(III) oxides. Media containing akaganeite (β-FeOOH), another form of insoluble Fe(III) commonly synthesized by slow NaOH addition to $FeCl_3$ solutions, was also used and showed similar growth trends but at a slower rate. Minimal medium containing 20 mM acetate as the electron donor and Fe(III) oxide or Fe(III) citrate as the sole electron acceptor was inoculated 1:100 from the acetate-fumarate grown culture. To monitor Fe(III) reduction over time, 0.1 mL of the Fe(III) medium was removed at regular intervals and dissolved in 0.9 mL of 0.5 N HCl for at least 24 hr in the dark. The acid extractable Fe(II) was measured using a modified FerroZine assay (*Chan et al., 2015*).

Three-electrode bioreactors with a working volume of 15 ml were assembled as previously described (*Marsili et al., 2008*). The potential of the polished graphite working electrode with a surface area of 3 $cm^2$ was maintained at −0.10 V vs. standard hydrogen electrode (SHE) using a VMP3 multichannel potentiostat (Biologic), a platinum counter electrode and calomel reference. This potential mimics Fe(III) oxides used in parallel experiments. Reactors were inoculated as previously described (*Chan et al., 2017*). Bioreactors were maintained at 30°C under a constant stream of humidified $N_2:CO_2$ (80:20) scrubbed free of oxygen by passage over a heated copper furnace. In prior work, the oxygen concentration in the headspace of these reactors has been shown to be ~1 ppm.

## Strain construction

The sucrose-SacB counter-selection strategy was used to generate a scarless *gacA* or *esnD* deletion strain (*Chan et al., 2015*). *Supplementary file 2* lists primers and restriction enzymes used to generate the ~750 bp flanking fragments of the *gacA* or *esnD* sequences to ligate into pK18mobsacB. The *E. coli* S17-1 donor strain mobilized plasmids into *G. sulfurreducens*. To integrate downstream of the *glmS* gene using Tn7 (*Damron et al., 2013*), derivatives of *E. coli* MFDpir carrying pTNS3 (encoding the Tn7 transposase TnsABCD) and MFDpir carrying a modified suicide vector pTJ1 with the sequence of interest cloned between the n7L and n7R sites was combined with *G. sulfurreducens* recipient strains by centrifugation in the anoxic glovebox, then incubation of the cell mixture on top of a filter paper disk (Millipore GPWP04700) placed on 1.5% agar with acetate-fumarate plates for 4 hr before plating on spectinomycin selective medium. Amplification and sequencing of the insertion junction revealed that TnsABCD mediated Tn7 integration is site specific in *G. sulfurreducens* and is 25 bp downstream of the *glmS* (GSU0270) stop codon.

## Plasmid and reporter construction

Plasmids expressing *gacA* and *gacA* site variants were cloned into pRK2-Geo2 (*Chan et al., 2015*) with the native promoter (replacing the *acpP* promoter) or over-expressed from the *acpP* promoter in pRK2-Geo2. The native *gacA* promoter was fused to *gacA* and *gacA* variants by extending two

oligos coding for the *gacA* promoter with *gacA* PCR fragments using overlap PCR. The cGAMP selective riboswitch controlling GSU1761 was fused to the Nanoluc gene with overlap PCR and cloned into pTOPO2.1. The luminescent reporter provided a strong signal even at low levels of expression compared to GFP, and allowed us to circumvent the problem of high autofluorescence in crude protein lysates due to the abundance of cytochromes in *G. sulfurreducen*s. The reporter was made by fusing the natural cGAMP-specific GSU1761 riboswitch upstream of the Nanoluc gene (pNL1.1, Promega). Analysis of the riboswitch expression platform suggests that cGAMP binding stabilizes anti-terminator formation and thus turns on expression of the Nanoluc reporter gene. The cGAMP selective GSU1761 riboswitch was replaced with a mutant Gmet_0970 riboswitch using Gibson assembly to generate a cdiG selective Nanoluc fusion. The cdiG selectivity of this mutant GEMM riboswitch is confirmed using gel-shift analysis (*Figure 2*). Tn7 integrative plasmids expressing *gacA*, *gacA* mutants, *esnD* and Nanoluc fusions were sub-cloned with either the native promoter or the *acpP* promoter into a derivative of pTJ1 (*Damron et al., 2013*) from the pRK2-Geo2 backbone by sequential digest with NheI and blunted with Klenow enzyme before digesting with AscI. Genes under the native promoter or *acpP* promoter were then ligated into the AscI and PmeI site into the pTJ1 derivative Tn7 integration plasmid. Cyclic dinucleotide Nanoluc fusion plasmids were subcloned into pRK2-Geo2 and Tn7 integrative plasmids to report cGAMP or cdiG levels in *G. sulfurreducens* strains.

For the crystallography construct, the T4 lysozyme sequence containing an E11Q inactivating mutation without stop codon was placed upstream of the Hypr GGDEF domain of Gmet_1914 (residues 294–459) sequence. This chimeric protein coding sequence was inserted between the NdeI and XhoI restriction sites of pET24a using restriction digest-ligation techniques. For in vitro analysis of mutants, site-directed-mutagenesis with the around-the-horn mutagenesis technique [https://openwetware.org/wiki/%27Round-the-horn_site-directed_mutagenesis] was used on a previously reported plasmid for expression of MBP-tagged R393A GSU1658 (*Hallberg et al., 2016*) to generate GacA mutant constructs. For WspR constructs used in flow cytometry assays, codon-optimized WspR (Key Resources Table) was inserted between the NdeI and XhoI restriction sites of pCOLA-Duet-1 using restriction digest-ligation techniques. This wild-type sequence was used as the template for round-the-horn mutagenesis. All primers and restriction enzyme used are listed in *Supplementary file 2*.

## Nanoluc assay

*G. sulfurreducens* strains with either the cdiG- or the cGAMP-Nanoluc reporter integrated into the Tn7 site were grown to mid-log fumarate-limited medium (40 mM acetate and 80 mM fumarate), the same ratio of acetate:fumarate as in RNA-seq conditions, and lysed at room temperature for 5 min in a phosphate-buffered saline (PBS) solution containing $1 \times$ BugBuster (Novagen) and 0.3 mg/ml DNase. 10 µl of the Nanoglo reagent (Promega) and 10 µl of the cell lysate were combined in a white-bottom, 96 well plate and luminescence was detected at 461 nm (Molecular Devices). Biological replicates (n = 3) were assayed. In assays where the luminescence exceeded the linear range of the spectrophotometer or deviated from steady-state, lysates were diluted in PBS before combining with the Nanoglo reagent.

## RNA-seq

Total RNA was extracted from 10 mL of *G. sulfurreducens* electron acceptor limited culture grown to mid-log (0.25–0.3 OD). Cell pellets were washed in RNAprotect (Qiagen) and frozen at −80°C before RNA extraction using RNeasy with on column DNase treatment (Qiagen). Ribosomal RNA was depleted using RiboZero (Illumina) by the University of Minnesota Genomics Center before stranded synthesis and sequenced on Illumina HiSeq 2500, 125 bp pair-ended mode. Residual ribosomal RNA sequences (<1%) were removed before analysis using Rockhopper, an RNAseq analysis program specifically designed to analyze bacterial transcriptomes (McClure et al., 2013). Duplicate biological samples were analyzed for each strain. Each replicate had between 13–14 M passing filter reads. Rockhopper aligned the rRNA depleted reads to our laboratory re-sequenced and re-annotated *G. sulfurreducens* genome, then normalized read counts from each experimental replicate by the upper quartile gene expression before they are compared. Raw reads and re-sequenced

genome data have been deposited to the NCBI SRA database PRJNA290373 (*Chan et al., 2015*). Full RNA-seq expression data are in *Supplementary file 1*.

## Overexpression and purification of dinucleotide cyclase enzymes

Full-length proteins with N-terminal $His_6$-MBP tags encoded in pET16-derived plasmids (cGAS and DncV plasmids are from (*Kranzusch et al., 2014*), WT GacA is from (*Hallberg et al., 2016*), and mutants are from this study) and the T4 lysozyme-$Gmet\_1914^{294-459}$ GGDEF chimera protein with C-terminal $His_6$ tag encoded in pET24a were overexpressed in *E. coli* BL21 (DE3) Star cells harboring the pRARE2 plasmid encoding human tRNAs (Novagen). Briefly, an aliquot of the overnight starter culture was re-inoculated into LB with antibiotics (LB/Carb/Chlor for pET16, LB/Kan/Chlor for pET24a) and grown to an $OD_{600}$ ~0.7, after which cultures were induced with 1 mM IPTG for 10 hr. After centrifugation to isolate the cell pellet, cells were lysed by sonication in lysis buffer (25 mM Tris-HCl (pH 8.2), 500 mM NaCl, 20 mM imidazole, and 5 mM beta-mercaptoethanol). Lysate was then clarified by centrifugation at $10,000 \times g$ for 45 min at 4°C. Clarified lysate was bound to Ni-NTA agarose (Qiagen), and resin was washed with $3 \times 20$ mL lysis buffer prior to elution with lysis buffer supplemented with 500 mM imidazole. Purified proteins were dialyzed overnight at 4°C into storage buffer (20 mM HEPES-KOH (pH 7.5), 250 mM KCl, 1 mM TCEP, and 5% (v/v) glycerol). Proteins purified in this way were concentrated to ~5–10 mg/mL, flash frozen in liquid nitrogen, and stored at −80°C. Protein purity was assessed by SDS-PAGE.

The crystallization fusion construct, T4lysozyme-$Gmet\_1914^{294-459}$-$His_6$, was further purified by size-exclusion chromatography on a Superdex 200 16/60 column in gel-filtration buffer (20 mM HEPES-KOH (pH 7.5), 250 mM KCl, 1 mM TCEP, and 5% (v/v) glycerol), and eluted protein was concentrated to 10 mg/mL. Purified protein was used immediately for x-ray crystallography or flash frozen in liquid nitrogen and stored at −80°C for biochemical experiments.

## In vitro activity assay of dinucleotide cyclases using radiolabeled NTPs

In vitro activity assays were performed as previously described (*Kranzusch et al., 2014*) as independent technical replicates (n = 3, assays used the same stock enzyme preparation in separate reaction mixtures), with the following modifications. Enzyme (10 µM) was incubated in reaction buffer (50 mM Tris-HCl (pH 7.5), 100 mM NaCl, 10 mM $MgCl_2$, and 5 mM dithiothreitol) with 100 µM each NTP substrate and/or nonhydrolyzable analog, and ~0.1 µCi radiolabeled $[\alpha\text{-}^{32}P]$-ATP or $[\alpha\text{-}^{32}P]$-GTP (Perkin Elmer) at 28°C for 1 hr. After the reaction, the mixture was treated with 20 U of Calf Intestinal Alkaline Phosphatase (NEB) at 28°C for 30 min to digest unincorporated NTPs, followed by heating to 95°C for 30 s to terminate the reaction. The reaction mixture (1 µL) was spotted onto a PEI-cellulose F thin layer chromatography plate (Millipore), and allowed to dry for 15 min at room-temperature. TLC plates were developed using 1 M $KH_2PO_4$ (pH 3.6) as the mobile phase. Plates were dried overnight and radiolabeled products were detected using a phosphor-imager screen (GE Healthcare) and Typhoon Trio +scanner (GE Healthcare).

## In vitro activity assay of dinucleotide cyclases using pyrophosphatase assay

In vitro activity assays were performed as previously described for diguanylate cyclases (*Burns et al., 2014*) as independent technical replicates (n = 3, assays used the same stock enzyme preparation in separate reaction mixtures), with the following modifications. The EnzChek pyrophosphate kit (Life Technologies) was used according to the manufacturer's instructions except the buffer was supplemented with KCl to a final concentration of 100 mM and $MgCl_2$ to a final concentration of 10 mM, and the reactions were initiated with addition of ATP or GTP. Assays were performed in triplicate in Corning Costar 96 well black, clear-bottomed plates containing 1 µM protein and varying NTP concentrations (0–10 mM). Absorbance at 360 nm in each well was measured using a SpectraMax i3x plate reader (Molecular Devices) and SoftMax Pro 6.5.1 software. Subsequent analyses to determine enzymatic rates were performed using the Excel Solver package.

## In vitro activity assay of dinucleotide cyclases using LC-MS

Activity assays with ATP, GTP, or mixtures of ATP and GTP were performed as described previously (*Burhenne and Kaever, 2013*; *Hallberg et al., 2016*) in independent technical replicates (n = 3, assays

used the same stock enzyme preparation in separate reaction mixtures). For unnatural substrates, the reactions were performed using 5 μM GacA and 200 μM of unnatural NTP at 37°C for 16–20 hr. Prior to LC-MS analysis, samples were treated with a 60°C incubation as in ref. (*Gentner et al., 2012*) to ensure all analyzed CDN samples were monomeric. LC-MS analysis of enzyme reactions was performed using an Agilent 1260 Quadrupole LC-MS with an Agilent 1260 Infinity HPLC equipped with a diode array detector. Sample volumes of 10 μL were separated on a Poroshell 120 EC C18 column (50 mm length ×4.6 mm internal diameter, 2.7 μm particle size, Agilent) at a flow rate of 0.4 mL/min. For analysis of enzyme reactions, an elution program consisting of 0% B for 5 min, followed by linear elution of 0% to 10% B over 1.5 min, isocratic elution at 10% B for 2 min, linear elution of 10% to 30% B over 2.5 min, linear elution from 30% to 0% B over 10 min, and isocratic elution of 0% B for 4 min, 50 s was used. Solvent A was 10 mM ammonium acetate/0.1% acetic acid and solvent B was HPLC-grade methanol. Under these conditions, the retention times are 10.23 ± 0.02 min for cdiG, 10.56 ± 0.02 min for cGAMP, and 11.09 ± 0.05 min for cdiA. The assignment of cyclic dinucleotide identity was confirmed through analysis of the mass spectra in the positive ion mode using m/z range = 150 to 1000. Product ratios were quantified using peak integrations at 254 nm by comparison to standard curves generated for each cyclic dinucleotide at known concentrations.

## Computational modeling of dinucleotide cyclase activity

We develop our mathematical derivation from the scheme presented in *Figure 5A*. This kinetic scheme assumes that the active dimer (E) remains at a constant total concentration throughout the reaction compared to inactive monomeric enzyme, which is not included in the model. The model also assumes that the rate-determining step is the production of the linear intermediate, which implies that the intermediate is efficiently converted to the cyclic dinucleotide. While the linear intermediate (pppGpG) is sometimes observed for diguanylate cyclases in vitro (*Skotnicka et al., 2016*), this may be due to the characterization of non-activated enzymes in vitro. We do not see significant buildup of any linear intermediates for in vitro or in vivo reactions with GacA, which supports our model.

With this assumption, our kinetic model contains 14 equations, describing the change in concentration of each relevant compound in the reaction:

$$d[E]/dt = k_{n1G}[EnG] - k_{1G}[E][G] + k_{n1G}[EGn] - k_{1G}[E][G] + k_{n1A}[EAn] - k_{1A}[E][A] + k_{n1A}[EnA] - k_{1A}[E][A] + k_{cat}[EGG] + k_{cat}[EGA] + k_{cat}[EAA] + k_{cat}[EAG]$$ (1)

$$d[EnG]/dt = k_{1G}[E][G] - k_{n1G}[EnG] - k_{2G}[EnG][G] + k_{n2G}[EGG] + k_{nA|G}[EAG] - k_{A|G}[EnG][A]$$ (2)

$$d[EGn]/dt = k_{1G}[E][G] - k_{n1G}[EGn] + k_{n2G}[EGG] - k_{2G}[EGn][G] + k_{nA|G}[EGA] - k_{A|G}[EGn][A]$$ (3)

$$d[EnA]/dt = k_{1A}[E][A] - k_{n1A}[EnA] + k_{nG|A}[EGA] - k_{G|A}[EnA][G] + k_{n2A}[EAA] - k_{2A}[EnA][A]$$ (4)

$$d[EAn]/dt = k_{1A}[E][A] - k_{n1A}[EAn] + k_{n2A}[EAA] - k_{2A}[EAn][A] + k_{nG|A}[EAG] - k_{G|A}[Ean][G]$$ (5)

$$d[EGG]/dt = k_{2G}[EnG][G] - k_{n2G}[EGG] + k_{2G}[EGn][G] - k_{n2G}[EGG] - k_{cat}[EGG]$$ (6)

$$d[EGA]/dt = k_{A|G}[EGn][A] - k_{nA|G}[EGA] + k_{G|A}[EnA]][G] - k_{nG|A}[EGA] - k_{cat}[EGA]$$ (7)

$$d[EAG]/dt = k_{A|G}[EnG][A] - k_{nA|G}[EAG] + k_{G|A}[EAn][G] - k_{nG|A}[EAG] - k_{cat}[EAG]$$ (8)

$$d[EAA]/dt = k_{2A}[EnA][A] - k_{n2A}[EAA] + k_{2A}[EAn][A] - k_{n2A}[EAA] - k_{cat}[EAA]$$ (9)

$$d[\text{cdiA}]/dt = \text{k}_{\text{cat,cdiA}}[\text{EAA}] \tag{10}$$

$$d[\text{cdiG}]/dt = \text{k}_{\text{cat,cdiG}}[\text{EGG}] \tag{11}$$

$$d[\text{cAG}]/dt = \text{k}_{\text{cat,cAG}}[\text{EAG}] + \text{k}_{\text{cat,cAG}}[\text{EGA}] \tag{12}$$

$$\begin{aligned} d[\text{A}]/dt = &-\text{k}_{1\text{A}}[\text{E}][\text{A}] + \text{k}_{\text{n1A}}[\text{EAn}] - \text{k1A}[\text{E}][\text{A}] + \text{k}_{\text{n1A}}[\text{EnA}] - \text{k}_{\text{A|G}}[\text{EGn}][\text{A}] + \text{k}_{\text{nA|G}}[\text{EGA}] - \\ &\text{k}_{2\text{A}}[\text{EnA}][\text{A}] + \text{k}_{\text{n2A}}[\text{EAA}] - \text{k}_{2\text{A}}[\text{EAn}][\text{A}] + \text{k}_{\text{n2A}}[\text{EAA}] - \text{k}_{\text{A|G}}[\text{EnG}][\text{A}] + \text{k}_{\text{nA|G}}[\text{EAG}] \end{aligned} \tag{13}$$

$$\begin{aligned} d[\text{G}]/dt = &-\text{k}_{1\text{G}}[\text{E}][\text{G}] + \text{k}_{\text{n1G}}[\text{EnG}] - \text{k}_{1\text{G}}[\text{E}][\text{G}] + \text{k}_{\text{n1G}}[\text{EGn}] - \text{k}_{2\text{G}}[\text{EnG}][\text{G}] + \text{k}_{\text{n2G}}[\text{EGG}] - \\ &\text{k}_{2\text{G}}[\text{EGn}][\text{G}] + \text{k}_{\text{n2G}}[\text{EGG}] - \text{k}_{\text{G|A}}[\text{EnA}][\text{G}] + \text{k}_{\text{nG|A}}[\text{EGA}] - \text{k}_{\text{G|A}}[\text{EAn}][\text{G}] + \text{k}_{\text{nG|A}}[\text{EAG}] \end{aligned} \tag{14}$$

Where each variable is:

E, active enzyme

A and G, ATP and GTP

EnG, enzyme with binding pocket 1 empty and binding pocket 2 with GTP

EGn, enzyme with binding pocket 1 with GTP, binding pocket 2 empty

EnA, as EnG, except with ATP

EAn, as EGn, except with ATP

EGG, enzyme with two GTP bound

EAA, enzyme with two ATP bound

EAG, enzyme with ATP in binding pocket 1, GTP in binding pocket 2

EGA, enzyme with GTP in binding pocket 1, ATP in binding pocket 2

And where reverse rate constants of a reaction are denoted by 'n' (i.e. $\text{k}_{1\text{G}}$ is the forward rate constant for the first GTP binding event, whereas $\text{k}_{\text{n1G}}$ is the rate constant for GTP dissociating). Thus, the equilibrium constant values shown in *Figure 3A* are related to on/off rates by the following equations:

$$\text{K}_{1\text{A}} = \text{k}_{\text{n1A}}/\text{k}_{1\text{A}} \tag{15}$$

$$\text{K}_{2\text{A}} = \text{k}_{\text{n2A}}/\text{k}_{2\text{A}} \tag{16}$$

$$\text{K}_{1\text{G}} = \text{k}_{\text{n1G}}/\text{k}_{1\text{G}} \tag{17}$$

$$\text{K}_{2\text{G}} = \text{k}_{\text{n2G}}/\text{k}_{2\text{G}} \tag{18}$$

$$\text{K}_{\text{A|G}} = \text{k}_{\text{nA|G}}/\text{k}_{\text{A|G}} \tag{19}$$

$$\text{K}_{\text{G|A}} = \text{k}_{\text{nG|A}}/\text{k}_{\text{G|A}} \tag{20}$$

For the single-substrate case, we utilize the exact solution provided by Oliveira *et al.* to obtain all dissociation constants for the first and second binding events, as well as the catalytic rate constant. Importantly, this only gives dissociation constants, which we convert to on and off rate constants using an arbitrary assignment of $\text{k}_{\text{no}}$. Because $\text{K}_{\text{D}} = \text{k}_{\text{off}}/\text{k}_{\text{on}}$, and because the $\text{k}_{\text{cat}}$ values are <<1 sec$^{-1}$, we arbitrarily set $\text{k}_{\text{on}}$ values to 1 µM$^{-1}$sec$^{-1}$. Thus, $\text{k}_{1\text{A}}, \text{k}_{2\text{A}}, \text{k}_{1\text{G}}, \text{k}_{2\text{G}}, \text{k}_{\text{A|G}}, \text{and} \, \text{k}_{\text{G|A}}$ are all set to 1 µM$^{-1}$sec$^{-1}$.

To calculate endpoint product ratios, we performed numerical integration (using the Python ODEint solver package in NumPy) of the system of differential equations using the starting concentrations over an hour-long time course — equivalent in length to our experimental procedures — using 1 s intervals. As stated in the main text, the value of $\text{k}_{\text{cat,AG}}$ was set conservatively to the same value as $\text{k}_{\text{cat,diG}}$ (0.03 sec$^{-1}$). The values for $\text{K}_{\text{A|G}}$ and $\text{K}_{\text{G|A}}$ were tested between 1 and 100 µM (corresponding to varying $\text{k}_{\text{no}}$ between 1 and 100 µM sec$^{-1}$) using a step size of 1 µM. Thus, for each

combination of $K_{A|G}$ and $K_{G|A}$ tested (10,000 possible combinations), we performed linear updates over 3,600 1 s steps of the 14 analytes, using the generic equation:

$$[\text{Analyte}]_{t+1} = [\text{Analyte}]_t + t*d[\text{Analyte}]/dt \tag{21}$$

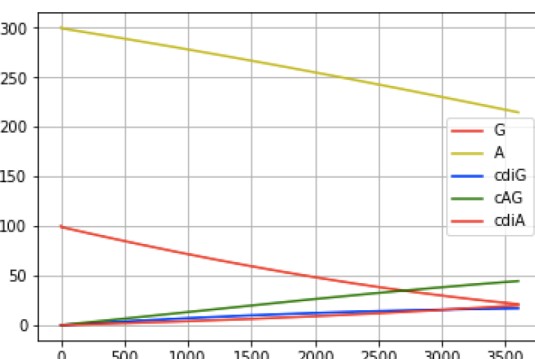

**Scheme 1.** Example reaction time course via simulation.
DOI: https://doi.org/10.7554/eLife.43959.023

Because we do not include any noise in these equations, the simulation gives the same result each time.

The ratio of each cyclic dinucleotide, which was calculated for the 1 hr endpoint, is:

$$\text{ratio CDN} = [\text{CDN}]_{1h}/([\text{cdiG}]_{1h} + [\text{cAG}]_{1h} + [\text{cdiA}]_{1h}) \tag{22}$$

We calculated the model error as the sum of the least squares difference between the experimental product ratios and modeled results (*Equation 22*) for each starting ATP/GTP ratio. The best fit values for $K_{A|G}$ and $K_{G|A}$ were the combination that gave the lowest model error (*Figure 3B*). Parameter values for the best fit kinetic model are shown below (also see *Table 1*):

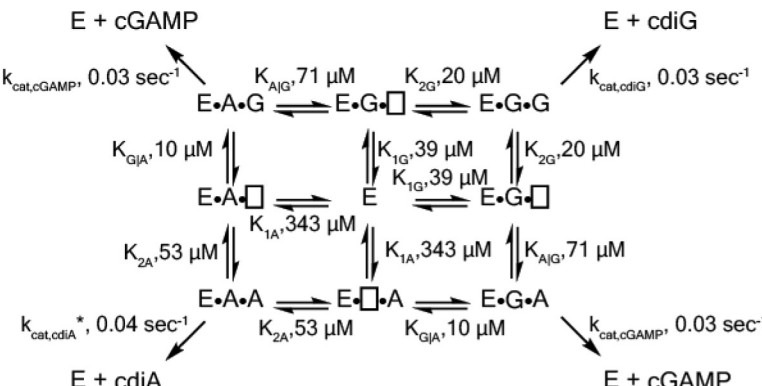

**Scheme 2.** Best fit kinetic model from combined experimental and simulation data.
DOI: https://doi.org/10.7554/eLife.43959.024

To model cellular homeostasis as shown in *Figure 5D and E*, the numerical integration program was run with $d[A]/dt$ and $d[G]/dt$ (*Equations 13 and 14*) set to zero.

## Liquid culture growth of *E. coli* BL21 (DE3) Star for Nucleotide Extraction

BL21 (DE3) Star cells containing the pRARE2 plasmid (Novagen) and pET24a plasmid encoding synthase enzyme constructs were inoculated into LB/kan/chlor at 37°C with shaking at 250 rpm overnight. An aliquot of the starter culture was re-inoculated into LB/kan/chlor media and grown to an $OD_{600}$ ~0.3, after which cultures were induced with 1 mM IPTG at 28°C for 4 hr. Cells were harvested by centrifugation at 4,700 rpm for 15 min at 4°C, and pellets were stored at −80°C.

## Cell extraction and LC-MS analysis of *E. coli*.

Cyclic dinucleotides were extracted as described previously (*Hallberg et al., 2016*) from two biological replicates. LC-MS analysis of *E. coli* cell extracts was performed as described previously (*Hallberg et al., 2016*).

## Crystallization and structure determination of T4Lys-Gmet_1914

Prior to crystallization, T4Lysozyme-Gmet_1914$^{294–459}$ protein was incubated for 10 min at rt at a concentration of 6.5 mg ml$^{-1}$ in the presence of 10 mM GTP and 10 mM MgCl$_2$. The T4Lysozyme-Gmet_1914$^{294–459}$–Guanosine nucleotide complex was crystallized in a hanging-drop vapor diffusion format using the final optimized crystallization conditions of 30 mM HEPES-KOH (pH 7.5), 300 mM Na(OAc), and 26% PEG-4000. Crystals were grown in Easy-Xtal 15-well trays (Qiagen) in 2 µl hanging drops with a 1:1 (protein:reservoir) ratio over 350 µl of reservoir solution. Crystals required incubation at 18°C for 2–4 days for complete growth, and then were transferred with a nylon loop to a new drop containing reservoir solution supplemented with 10% glycerol as a cryoprotectant and incubated for 30 s before flash-freezing in liquid nitrogen. Native and anomalous data were collected under cryogenic conditions at the Lawrence Berkeley National Laboratory Advanced Light Source (Beamline 8.3.1).

X-ray diffraction data were processed with XDS and AIMLESS (*Kabsch, 2010*) in the monoclinic spacegroup *C* 2. Phase information was determined with a combination of molecular replacement and sulfur single-wavelength anomalous dispersion (SAD). Briefly, iterative sulfur-SAD data sets were collected at ~7,235 eV and merged from independent portions of a large T4Lysozyme-Gmet_1914$^{294–459}$ crystal as previously described (*Lee et al., 2016a*). A minimal core of T4-Lysozyme (PDB 5JWS) (*Lee et al., 2016b*) was used a search model for molecular replacement and sub-structure determination. The placed T4-Lysozyme fragment was then used to guide SAD identification of 17 sites with HySS in PHENIX (*Adams et al., 2010*) corresponding to 12 sulfur atoms in T4Lysozyme-Gmet_1914$^{294–459}$ and 5 solvent ion positions. SOLVE/RESOLVE (*Terwilliger, 1999*) was used to extend phases to the native T4Lysozyme-Gmet_1914$^{294–459}$ data processed to ~1.35 Å and model building and refinement were completed with Coot (*Emsley and Cowtan, 2004*) and PHENIX (*Table 2*).

## Bioinformatic analysis of GGDEF variants

A Python-based program was developed to extract alignment data for a library of 139,801 putative GGDEF domain-containing proteins from the Uniprot database (obtained through Pfam, accession PF00990, http://pfam.xfam.org/, accessed 06/05/2014). In particular, positions critical for catalytic activity (i.e. the GG[D/E]EF sequence) and selectivity (i.e. positions 347 and 303 in *Gs*GacA) were identified and analyzed for each sequence. Given previous results with some DGCs possessing altered signature motifs, we assigned any diguanylate cyclase with a [G/A/S]G[D/E][F/Y] motif to be active.

# Acknowledgments

X-ray data were collected at Beamline 8.3.1 of the Lawrence Berkeley National Lab Advanced Light Source (ALS). We thank Professors Robert Bergman and Jennifer Doudna for helpful comments and discussion, Jason Peters for providing the Tn7 vectors and Jonathan Badalamenti for sequence analysis assistance and RNA-seq data management. This work was supported by NSF MCB collaborative grants 1716256 and 1714196 (to MCH and DRB), ONR grant N000141612194 (to DRB), NIH R01 GM124589 (to MCH), and NSF graduate fellowship (to ZFH).

# Additional information

## Funding

| Funder | Grant reference number | Author |
|---|---|---|
| National Science Foundation | 1716256 | Zachary F Hallberg<br>Todd A Wright<br>Ming C Hammond |

| Office of Naval Research | N000141612194 | Chi Ho Chan<br>Daniel R Bond |
| National Institutes of Health | R01 GM124589 | Zachary F Hallberg<br>James J Park<br>Ming C Hammond |
| National Science Foundation | 1714196 | Chi Ho Chan |
| National Science Foundation | 1915466 | Todd A Wright<br>Ming C Hammond |

The funders had no role in study design, data collection and interpretation, or the decision to submit the work for publication.

### Author contributions
Zachary F Hallberg, Data curation, Formal analysis, Visualization, Methodology, Writing—original draft, Writing—review and editing; Chi Ho Chan, Formal analysis, Investigation, Visualization, Methodology, Writing—original draft, Writing—review and editing; Todd A Wright, Investigation, Methodology, Writing—review and editing; Philip J Kranzusch, Formal analysis, Supervision, Writing—review and editing; Kevin W Doxzen, Formal analysis, Writing—review and editing; James J Park, Investigation, Writing—review and editing; Daniel R Bond, Ming C Hammond, Conceptualization, Supervision, Funding acquisition, Visualization, Writing—original draft, Project administration, Writing—review and editing

### Author ORCIDs
Chi Ho Chan ⓘ http://orcid.org/0000-0002-6596-3436
Daniel R Bond ⓘ https://orcid.org/0000-0001-8083-7107
Ming C Hammond ⓘ http://orcid.org/0000-0003-2666-4764

### Decision letter and Author response
Decision letter https://doi.org/10.7554/eLife.43959.035
Author response https://doi.org/10.7554/eLife.43959.036

## Additional files

### Supplementary files
• Supplementary file 1 RNAseq. Raw RNAseq fastq reads from duplicate cultures of each strain was analyzed using Rockhopper (see methods). The re-sequenced and re-annotated reference genome was from our previously published work (*Chan et al., 2017*). Instead of normalizing the expression to the total number of reads expressed more commonly as reads per kilobase mapped, the expression value is normalized by Rockhopper for each sample to the upper quartile gene expression level, excluding genes with zero expression. This method of normalization better correlates RNAseq expression with qRT-PCR data in a separate test study. q-values are calculated from p-values to control the false discovery rate and used to indicate significance of differentially expression between conditions. A q-value of <0.01 is consider to be significant.

DOI: https://doi.org/10.7554/eLife.43959.025
• Supplementary file 2. Primers used in this study. Primers are listed from 5' - 3'
DOI: https://doi.org/10.7554/eLife.43959.026
• Transparent reporting form
DOI: https://doi.org/10.7554/eLife.43959.027

### Data availability
Raw RNA-seq reads and re-sequenced genome data have been deposited to the NCBI SRA database PRJNA290373. Diffraction data have been deposited in PDB under accession number 6D9M. All data generated or analysed during this study are included in the manuscript and supporting files. Source data files have been provided for Figures 1-4.

The following datasets were generated:

| Author(s) | Year | Dataset title | Dataset URL | Database and Identifier |
|---|---|---|---|---|
| Hallberg ZF, Chan CH | 2018 | Geobacter sulfurreducens MN1 genomic and expression studies | https://www.ncbi.nlm.nih.gov/bioproject/PRJNA290373 | NCBI BioProject, PRJNA290373 |
| Hallberg Z, Doxzen K, Kranzusch P, Hammond M | 2019 | T4-Lysozyme fusion to Geobacter GGDEF | https://www.rcsb.org/structure/6D9M | PDB, 6D9M |

The following previously published dataset was used:

| Author(s) | Year | Dataset title | Dataset URL | Database and Identifier |
|---|---|---|---|---|
| Chan CH, Levar CE, Jimenez-Otero F, Bond DR | 2017 | Geobacter sulfurreducens MN1 genomic and expression studies | https://www.ncbi.nlm.nih.gov/bioproject/PRJNA290373 | NCBI BioProject, PRJNA290373 |

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
