## [Decision Letter]

Thank you for submitting your article "cGAMP signaling controls a transient surface-associated lifestyle" for consideration by *eLife*. Your article has been reviewed by three peer reviewers, and the evaluation has been overseen by a Guest Editor and Gisela Storz as the Senior Editor. The following individual involved in the review of your submission has agreed to reveal his identity: Jacob G Malone (Reviewer #2).

Three experts and two editors have reviewed your manuscript, and discussed our decision. Based on the reviews and discussion, we ask that you revise your manuscript.

Summary

In this manuscript, Hallberg et al. explore the cGAMP signaling pathway of *Geobacter*. This work is highly impactful and novel for a number of reasons. It is the first study to demonstrate a physiological role for cGAMP in this bacterium, which had been previously speculated by the discovery of cGAMP-binding riboswitches. Interestingly, cyclic di-GMP (cdiG) and cGAMP appear to play opposing roles in either driving biofilm formation or iron reduction in solution, respectively. Additionally, the kinetic modeling and structural studies provide an explanation as to how the cGAMP synthase, GacA, which forms a homodimer can generate a mixed nucleotide product. Their findings have important implications on the potential evolutionary pathways for cdiG signaling that can lead to the generation of novel signals.

Major revisions

1) Reviewer #1 made several significant points regarding how the structural data is presented and interpreted. Please address each of the following points explicitly in the revised manuscript and the rebuttal, as the points have important implications for your proposed model and the interpretation of the data.

a) Subsection “X-ray crystal structure of Hypr GGDEF domain of *G. metallireducens* GacA bound to guanosine substrate”, structure of GacA's Hypr GGDEF domain bound to nucleotide: A structure of the GGDEF domain of GacA is presented bound to nucleotides. From the active site electron density shown in Figure 7, it seems the alpha phosphate is not resolved or not present, and only guanosine (plus pyrophosphate) was modeled. Much of the results focus on this active site coordinating guanosine and pyrophosphate in the crystal. Presumably, it is a result of GTP hydrolysis to GMP and pyrophosphate during the crystallization process although it is not clear why the alpha phosphate is not visible. GTP hydrolysis can be considered an off-pathways since no phosphodiester bond is formed. However, in the main figures, the authors chose to model GTP to depict a potential substrate pose, and much of the mechanistic discussion regarding substrate specificity and conservation of the active site in Hypr GGDEF vs. canonical GGDEF domains relies on this model. As part of this analysis, the Hypr GGDEF*guanosine*pyrophosphate structure was compared to a bona fide GGDEF*GTP complex. The structural arguments concerning distinct substrate preferences of the two enzymes are based on distance differences between the protein and nucleotide of less than an Ångstrom between the two complexes. Based on the available data, it is difficult to assess if the differences in nucleotide coordination originate from the artificial cleavage of the substrate, or – as the authors argue – from intrinsic difference in GTP coordination between the two enzyme sub-families, which could explain their distinct catalytic characteristics. This point is somewhat mitigated by the bioinformatic analysis of the GGDEF domain superfamily and mutational validation of the model. However, the authors should use caution when describing the crystal structure presented here. For example, at one point, it is argued that the Hypr GGDEF structure extends the short list of substrate-bound GGDEF proteins determined to date. While it is correct that only very few GGDEF protein structures were solved bound to substrate, it seems incorrect to discuss the Hypr GGDEF domain structure as one of them given that it is bound to a hydrolysed side product. Finally, the structure of substrate-bound PleD, a canonical GGDEF protein, was determined using a non-hydrolyzable analog (GTPalphaS; Wassmann et al., 2007). If attainable, a crystal structure of a GacA*GTPalphaS complex could provide the proper state for a comparative study with other substrate-bound GGDEF domains.

b) Structure presentation in main text: In general, the authors chose to show modeled structures in the main text to illustrate the proposed substrate bound state (see previous point) and predict the function of conserved active site residues. In order to emphasize experimental data over models and to avoid confusion, the crystallographically determined states should be shown in the main figures (alongside with these models).

c) Arguments regarding transient adhesion controlled by cGAMP: The title does not correspond well with the main conclusions that can be drawn from the experiments described in the manuscript. Specifically, the evidence supporting a role of cGAMP in controlling transient surface interaction appears rather indirect. The argument is based on the fact that colonization of electrodes by *Geobacter sulfurreducens* is usually associated with biofilm formation and cdiG signaling, whereas GacA facilitates interactions of the bacterium with Fe(III) particles through its production of cGAMP (without a major impact on biofilm formation of electrodes). Although cdiG- and cGAMP-producing enzymes have inverse effects on these two phenotypes (both in knock-out and overexpression studies), interaction kinetics between bacterial cells and Fe(III) particles were not assessed directly. It is possible that the RNA-seq data provides clues to implicate cellular processes usually associated with more transient interactions of bacteria with substrates, however such a discussion is missing in the present version of the manuscript. Furthermore, one cannot exclude mechanisms other than adhesion kinetics for how cGAMP may contribute to metal reduction specifically. Without further corroboration, it seems appropriate to adjust the title (and main text) to better represent the experimentally verified conclusions. Such modifications would not take away from the impact of this study, which was perceived as high.

2) All three reviewers felt that it was important to provide evidence for the role of the I-Site and its function. A couple of experiments were suggested along these lines: (i) purification of the enzyme to see if cdiG was indeed associated with the I-site (purifying GacA (and an inhibitory-site mutant) and analyzing the nucleotides that copurify with the protein – ideally, wild-type and the *esnD*-deletion strains would be used as expression hosts, but GacA expression in *E. coli* in the absence and presence of an active diguanylate cyclase may be a more accessible model system), and (ii) building an I-site mutant of GacA and assessing its effect. While including both of these experiments would be outstanding, either of the two experiments would speak to the fact that the I-site does indeed bind cdiG, and additionally, the genetic experiment would provide direct functional data for the I-site.

3) All three reviewers liked the kinetic model, although reviewer #3 felt that some additional data should be provided to support the model. We came to a consensus that the model should be included, but some text added to the Discussion highlighting which parts of the model need to be supported by additional experimentation in the future. In particular, it should be made clear that as long as the aspect of the model regarding the differential activation of the N-terminal domain is stated as a hypothesis that has arisen from the model and that this aspect of that model needs to be further explored in the future. We felt that this was an excellent opportunity to help drive the direction of the field.

---

## [Author Response]

Major revisions1) Reviewer #1 made several significant points regarding how the structural data is presented and interpreted. Please address each of the following points explicitly in the revised manuscript and the rebuttal, as the points have important implications for your proposed model and the interpretation of the data.a) Subsection “X-ray crystal structure of Hypr GGDEF domain of G. metallireducens GacA bound to guanosine substrate”, structure of GacA’s Hypr GGDEF domain bound to nucleotide: A structure of the GGDEF domain of GacA is presented bound to nucleotides. From the active site electron density shown in Figure 7, it seems the alpha phosphate is not resolved or not present, and only guanosine (plus pyrophosphate) was modeled. Much of the results focus on this active site coordinating guanosine and pyrophosphate in the crystal. Presumably, it is a result of GTP hydrolysis to GMP and pyrophosphate during the crystallization process although it is not clear why the alpha phosphate is not visible. GTP hydrolysis can be considered an off-pathways since no phosphodiester bond is formed. However, in the main figures, the authors chose to model GTP to depict a potential substrate pose, and much of the mechanistic discussion regarding substrate specificity and conservation of the active site in Hypr GGDEF vs. canonical GGDEF domains relies on this model. As part of this analysis, the Hypr GGDEF*guanosine*pyrophosphate structure was compared to a bona fide GGDEF*GTP complex. The structural arguments concerning distinct substrate preferences of the two enzymes are based on distance differences between the protein and nucleotide of less than an Ångstrom between the two complexes. Based on the available data, it is difficult to assess if the differences in nucleotide coordination originate from the artificial cleavage of the substrate, or – as the authors argue – from intrinsic difference in GTP coordination between the two enzyme sub-families, which could explain their distinct catalytic characteristics. This point is somewhat mitigated by the bioinformatic analysis of the GGDEF domain superfamily and mutational validation of the model. However, the authors should use caution when describing the crystal structure presented here. For example, at one point, it is argued that the Hypr GGDEF structure extends the short list of substrate-bound GGDEF proteins determined to date. While it is correct that only very few GGDEF protein structures were solved bound to substrate, it seems incorrect to discuss the Hypr GGDEF domain structure as one of them given that it is bound to a hydrolysed side product. Finally, the structure of substrate-bound PleD, a canonical GGDEF protein, was determined using a non-hydrolyzable analog (GTPalphaS; Wassmann et al., 2007). If attainable, a crystal structure of a GacA*GTPalphaS complex could provide the proper state for a comparative study with other substrate-bound GGDEF domains.

We sincerely apologize for the inaccurate statement regarding the Hypr GGDEF structure being a substrate-bound structure, this was an oversight that we are glad to correct. The claim has been deleted.

To make things more clear (for readers and ourselves), we have included in the main Figure 6 the original structure with guanosine:PPi bound alongside the structure with GTP modeled based on partial localized density for the alpha phosphate. We added to Figure 8 the Fo-Fc omit map of electron density contoured at 2.0 σ for the bound guanosine nucleotide.

We also revised the main text throughout to discuss the structure with “guanosine” instead of “GTP”. For example:

“For the latter, we were only able to find partial localized electron density for the alpha phosphate (Figure 8). It is likely that GTP was hydrolyzed during crystallization but remained coordinated in the active site with density now visible for the guanosine and beta-gamma pyrophosphate (PP_i_). For clarity, we show both the original and modeled structures with the alpha phosphate. The final guanosine nucleotide binds at the T4 lysozymeGGDEF interface and may act to stabilize the construct in a way that ATP cannot.”

Finally, we kept the discussion of the GGDEF versus GGEEF structures, but added the appropriate caveats. Also, the text was edited to make clear that the structural analysis led to a hypothesis (rather than a conclusion) that then was tested by mutagenesis and bioinformatics analysis:

“Overlaying Hypr and canonical GGDEF domains provides a potential explanation for these earlier results (Figure 6B), although an important caveat to this analysis is that the Hypr GGDEF structure contains guanosine plus PP_i_ bound instead of intact GTP in the case of the canonical GGDEF. […]To test whether the Mg^2+^-GGDEF interaction is indeed critical to GacA activity, we made the D374*E mutant of *Gs*GacA, which converts the motif to GGEEF (the * indicates that the numbering used corresponds to the *G. metallireducens* GacA structure, because *Gs*GacA is shorter by one amino acid).”

b) Structure presentation in main text: In general, the authors chose to show modeled structures in the main text to illustrate the proposed substrate bound state (see previous point) and predict the function of conserved active site residues. In order to emphasize experimental data over models and to avoid confusion, the crystallographically determined states should be shown in the main figures (alongside with these models).

We now include the original structure with guanosine:Ppi bound alongside the structure with GTP modeled based on partial localized density for the alpha phosphate in main Figure 6. We also include the modeled dimer showing the original, crystallized Tyr conformation in the monomer alongside the model showing with rotated Tyr. Because the number of figure panels is much expanded, we have broken up the original figure into two figures. The new Figure 6 focuses on interactions observed in the monomer, whereas the new Figure 9 focuses on crossdimer interactions.

c) Arguments regarding transient adhesion controlled by cGAMP: The title does not correspond well with the main conclusions that can be drawn from the experiments described in the manuscript. Specifically, the evidence supporting a role of cGAMP in controlling transient surface interaction appears rather indirect. The argument is based on the fact that colonization of electrodes by Geobacter sulfurreducens is usually associated with biofilm formation and cdiG signaling, whereas GacA facilitates interactions of the bacterium with Fe(III) particles through its production of cGAMP (without a major impact on biofilm formation of electrodes). Although cdiG- and cGAMP-producing enzymes have inverse effects on these two phenotypes (both in knock-out and overexpression studies), interaction kinetics between bacterial cells and Fe(III) particles were not assessed directly. It is possible that the RNA-seq data provides clues to implicate cellular processes usually associated with more transient interactions of bacteria with substrates, however such a discussion is missing in the present version of the manuscript. Furthermore, one cannot exclude mechanisms other than adhesion kinetics for how cGAMP may contribute to metal reduction specifically. Without further corroboration, it seems appropriate to adjust the title (and main text) to better represent the experimentally verified conclusions. Such modifications would not take away from the impact of this study, which was perceived as high.

Thank you for the feedback. We have changed the title to “Structure and mechanism of a Hypr GGDEF enzyme that activates cGAMP signaling to control extracellular metal respiration”

2) All three reviewers felt that it was important to provide evidence for the role of the I-Site and its function. A couple of experiments were suggested along these lines: (i) purification of the enzyme to see if cdiG was indeed associated with the I-site (purifying GacA (and an inhibitory-site mutant) and analyzing the nucleotides that copurify with the protein – ideally, wild-type and the esnD-deletion strains would be used as expression hosts, but GacA expression in E. coli in the absence and presence of an active diguanylate cyclase may be a more accessible model system), and (ii) building an I-site mutant of GacA and assessing its effect. While including both of these experiments would be outstanding, either of the two experiments would speak to the fact that the I-site does indeed bind cdiG, and additionally, the genetic experiment would provide direct functional data for the I-site.

The point about I-site function indeed is important. Experiment (i) has been done previously, so we have added a sentence to describe the results (subsection “GacA is essential for production of intracellular cyclic GMP-AMP (cGAMP)”):

“A second mechanism regulating canonical GGDEF domains involves a conserved allosteric inhibitory site (I-site) that binds cyclic dinucleotides. […] In extracts of cells overexpressing GacA, the major CDN present is cGAMP (Hallberg et al., 2016), yet GacA still purifies with bound cdiG, which supports the hypothesis that the I-site is specific for cdiG.”

In Figure 1D, we show the effect of complementation with GacA R393A (I-site) mutant on the riboswitch-luciferase reporter and in Figure 1E, we show the effect of complementation with GacA R393A (I-site) mutant on Fe(III) oxide reduction. These experiments seem similar to what the reviewers propose for experiment (ii).

3) All three reviewers liked the kinetic model, although reviewer #3 felt that some additional data should be provided to support the model. We came to a consensus that the model should be included, but some text added to the Discussion highlighting which parts of the model need to be supported by additional experimentation in the future. In particular, it should be made clear that as long as the aspect of the model regarding the differential activation of the N-terminal domain is stated as a hypothesis that has arisen from the model and that this aspect of that model needs to be further explored in the future. We felt that this was an excellent opportunity to help drive the direction of the field.

Thank you for the feedback. In particular, we have made clearer that differential activation is a hypothesis that comes out from the model by labeling the Figure 3E as “Proposed asymm. activation” and making the following text changes:

“While uniform effects on *kcat* values would not change product ratios, we used the kinetic model to simulate product ratios if *k*cat,AG was increased asymmetrically by activation more than *k*cat,diG or *k*cat,diA (Figure 5E). […] Taken together, these results show how cooperative binding, including selective cooperativity induced by the first substrate bound,and tuning substrate affinities to cellular concentrations, could make GacA predominantly produce cGAMP.”

We also moved the mechanistic hypothesis for asymmetric activation to the Discussion.